# X-ray Structure Determination, Antioxidant Voltammetry Studies of Butein and 2′,4′-Dihydroxy-3,4-dimethoxychalcone. Computational Studies of 4 Structurally Related 2′,4′-diOH Chalcones to Examine Their Antimalarial Activity by Binding to Falcipain-2

**DOI:** 10.3390/molecules26216511

**Published:** 2021-10-28

**Authors:** Ijeoma Okoye, Sandra Yu, Francesco Caruso, Miriam Rossi

**Affiliations:** Department of Chemistry, Vassar College, Poughkeepsie, NY 12604, USA; iokoye@vassar.edu (I.O.); sandrayu@vassar.edu (S.Y.); caruso@vassar.edu (F.C.)

**Keywords:** falcipain-2, antioxidant, chalcone, butein, superoxide

## Abstract

Malaria is a huge global health burden with resistance to currently available medicines resulting in the search for newer antimalarial compounds from traditional medicinal plants in malaria-endemic regions. Previous studies on two chalcones, homobutein and 5-prenylbutein, present in *E. abyssinica*, have shown moderate antiplasmodial activity. Here, we describe results from experimental and computational investigations of four structurally related chalcones, butein, 2′,4′-dihydroxy-3,4-dimethoxychalcone (DHDM), homobutein and 5-prenylbutein to elucidate possible molecular mechanisms by which these compounds clear malaria parasites. The crystal structures of butein and DHDM show that butein engages in more hydrogen bonding and consequently, more intermolecular interactions than DHDM. Rotating ring-disk electrode (RRDE) voltammetry results show that butein has a higher antioxidant activity towards the superoxide radical anion compared to DHDM. Computational docking experiments were conducted to examine the inhibitory potential of all four compounds on falcipain-2, a cysteine protease that is involved in the degradation of hemoglobin in plasmodium-infected red blood cells of the host. Overall, this work suggests butein as a better antimalarial compound due to its structural features which allow it to have greater intermolecular interactions, higher antioxidant activity and to create a covalent complex at the active site of falcipain-2.

## 1. Introduction

Malaria is an infectious disease that is caused by *Plasmodium spp.* protozoan parasites and transmitted via the bites of infected female *Anopheles* mosquitoes [1]. Since malaria parasites were first discovered in the blood of humans in 1880 by Charles Laveran, malaria has remained a significant global health challenge [1]. About 300 to 500 million people are infected with malaria every year but ultimately, 1 to 3 million people worldwide die from *Plasmodium falciparum* malarial infections annually. There are several factors affecting one’s susceptibility to fatal forms of malaria including biological, environmental, and societal factors. However, children below the age of five in sub-Saharan Africa typically constitute the most vulnerable population [2]. In 2019, approximately 94% of malaria cases and deaths worldwide occurred in Africa with Nigeria accounting for about 27% of the global malaria incidence [3]. Additionally, the 2020 Global Malaria Report from the World Health Organization revealed that about 50% of the total world population is at risk of malaria [3]. Hence, malaria is not just a tropical disease that disproportionately affects (sub) tropical countries, it is a severe global and public health crisis.

The first effective cure for malaria was discovered in the early 17th century in infusions of the bark of the *Cinchona officinalis* tree [4] with the active antimalarial components of the bark being quinine and cinchonine. Currently, there are several antimalarial drugs that are typically prescribed to individuals suffering from malaria including chloroquine, quinine, artemisinin, and artemisinin combination therapies (ACTs). Remarkably, most of these effective antimalarial drugs are derived from traditional medicinal plants/herbs that are usually local to malaria-endemic regions.

However, scientists have begun to observe the emergence of Artemisinin-resistant malaria parasites [5], thus increasing the need for novel antimalarial treatment options. Reliance on traditional herbs to treat malaria is more common in developing countries where the majority of the population is unable to access or afford pharmaceutical antimalarial drugs [6,7]. In certain countries in sub-Saharan Africa such as Nigeria, traditional and cultural-based products such as palm wine, fermented foods like pap (*ogi*), and naturally occurring edible herbs are some of the most common local remedies for malaria [8,9,10]. Specifically, infusions of traditional medicinal plants are often prepared and consumed to treat malaria infections. Examples of such traditional medicinal plants include *Erythrina abyssinica* (coral tree, Uganda coral, or lucky-bean tree), *Azadirachta indica* (neem Tree), and *Moringa oleifera* (drumstick tree) [11].

In particular, *Erythrina abyssinica,* a leguminous tree that is native to East Africa but is also found in Central and South Africa, has been comprehensively studied and shown to contain bioactive compounds which confer to the plant its many therapeutic properties such as its moderate to high antimicrobial and antimalarial activities [12]. The most frequently used medicinal parts of the *E. abyssinica* plant are the stem bark and the roots, which are usually consumed via infusions, powder, pastes, or direct chewing [12]. Herbal preparations from the *E. abyssinica* plant are used to treat a wide variety of ailments including cancer, tuberculosis, HIV/AIDS, obesity, Type II diabetes, and malaria amongst many others [12]. Specifically, the extract from the stem bark of *E. abyssinica* has been demonstrated to have moderate to high antimalarial activity due to the antiplasmodial properties of two of its chemical constituents, 5-Prenylbutein and Homobutein [13,14].

Homobutein (**3**) and 5-Prenylbutein (**4**) are chalcones, frequently produced as a part of the plant’s defense system protecting against exposure to ultraviolet rays, pathogens, and toxins through their anti-inflammatory and high antioxidant activity [15,16]. Due to their simple chemical structure and relatively low cost of synthesis, chalcones are very attractive scaffolds that are used in medicinal chemistry as promising starting points and templates for developing effective and affordable drugs for infectious diseases. This research project focuses on investigating the antimalarial potential of 4 structurally related 2′,4′-diOH chalcones, Butein (**1**), 2′,4′-dihydroxy-3,4-dimethoxychalcone (**2**, DHDM), Homobutein (**3**) and 5-Prenylbutein (**4**), shown in Figure 1. Butein is a chalcone isolated from natural plants such as *Toxicodendron vernicifluum (Rhus verniciflua), Butea monosperma, Semecarpus anacardium, Dalbergia odorifera* while DHDM is a synthesized chalcone, differing from Butein only in its substitution of two hydroxyl groups with two methoxy groups [16]. Currently, the antimalarial potential of DHDM has not been investigated. However, Butein has been demonstrated to have high antiplasmodial activity due to its iron chelation and high antioxidant properties [17].

Among other natural products, chalcones have been studied for their antimalarial activity in the quest to understand their mechanism of action [18,19,20,21,22]. These earlier studies point to the ability of chalcones to inhibit the *P. falciparum* cysteine protease, falcipain-2, important in the hemoglobin degradation pathway, without which *Plasmodium* parasites cannot grow and survive in infected red blood cells [23]. An aim of this research project was to identify structural aspects useful for antimalaria activity found in the four 2′,4′-diOH chalcones described earlier. To do this, we characterized Butein and DHDM utilizing single crystal X-ray diffraction and measured their antioxidant activity since these compounds are commercially available. We also propose additional mechanisms by which the compounds **1**–**4** in Figure 1 could have an antimalarial effect via their antioxidant capacity and their inhibitory activity with Falcipain-2. We used Biovia Discovery Studio software to perform docking simulations and to examine the binding ability of compounds **1**–**4** to the active site of Falcipain-2. Then, we performed Rotating Ring Disk Electrode (RRDE) voltammetry experiments to ultimately determine the antioxidant activity of Butein (**1**) and DHDM (**2**) by measuring their rate of scavenging generated superoxide radicals. As *Plasmodium* parasites degrade hemoglobin to produce heme in the red blood cells of their host, they generate high levels of reactive oxygen species (ROS) which are toxic to the human cells. Hence, compounds with high antioxidant activity serve as effective antimalarial agents and abate the progress of a malarial infection by trapping toxic ROS such as superoxide radicals generated by malaria parasites during its hemoglobin degradation pathway [24].

## 2. Materials and Methods

### 2.1. Materials

Butein and DHDM were obtained from Cayman Chemical Company (Ann Arbor, MI, USA) and Indofine Chemical Company (Hillsborough, NJ, USA), US, respectively. Butein was kept in the freezer while DHDM was kept at room temperature

### 2.2. Single Crystal X-ray Diffraction and Analysis

An APEX2 DUO platform X-ray diffractometer from Bruker Advanced X-ray Solutions was used to obtain X-ray data measurements at 125 K; these were analyzed to determine the 3D crystal structures of both compounds. Crystal structures were solved using ShelX programs [25]. Appropriate crystals of DHDM for obtaining a high-resolution diffraction pattern were obtained from methanol solution. Butein crystals were obtained from a 1:1 methanol/ethanol solvent mixture that had been placed in the freezer at 4 °C to allow minimal exposure to light and air. Further analysis of the diffraction results was performed to identify the structural properties of both compounds including their intra and inter-molecular interactions. Butein and DHDM CIF files have been deposited at Cambridge Structural Database CSD (2092939 and 2092940).

### 2.3. RRDE Measurement of Antioxidant Activity

Materials used to determine the antioxidant activity of Butein and DHDM were tetrabutylammonium bromide (TBAB; Sigma Aldrich, St. Louis, MO, USA), 99% anhydrous Dimethyl Sulfoxide (DMSO; Sigma Aldrich), DHDM, and Butein. The 0.1 M TBAB/DMSO solution was used to produce electric current and enhance the occurrence of redox reactions via the gain and loss of electrons. Antioxidant activity was measured via the hydrodynamic voltammetry technique with a rotating ring disk electrode (RRDE). The equipment used in this experiment was an MSR electrode rotator with CE and ETL marks, together with a WaveDriver 20 benchtop USB Galvanostat from Pine Instrumentation. The main electrode tip was an E6RI ChangeDisk with a rigid gold ring and gold disk (Au/Au) insert. Before and after each experiment, 0.3 μL Alumina suspension was used to clean the main electrode tip. A platinum (Pt) reference electrode (yellow electrode) and Pt counter electrode (green electrode) were also used in this experiment. All electrodes were obtained from Pine Instrumentation.

The antioxidant activity of DHDM and Butein was determined individually based on the superoxide radical scavenging ability of each compound that was measured using the protocol developed in our lab [26]. Stock solutions of 0.025 M of DHDM in DMSO and 0.029 M of Butein in anhydrous DMSO were used in trials. For the experiment, the electrolytic cell was bubbled for 5 min with a dry O_2_/N_2_ (35%/65%) gas mixture to establish its dissolved oxygen level. The Au disk electrode was then rotated at 1000 rpm while the disk was swept from 0.2 V to −1.2 Volts and the ring was held constant at 0.25 Volts, the disk voltage sweep rate was set to 25 mV/s.

In summary, 7 runs were performed in the RRDE experiment to determine the antioxidant activity with 10, 20, 40, 80 and 160 of DHDM. For Butein, 8 runs were performed, from 10 to 160 μL (plus an additional 320 μL).

Results from each run were collected on Aftermath software and represented as voltammograms showing current-density vs. potential graphs that were later analyzed using Microsoft Excel. In an RRDE voltammetry experiment, the generation of the superoxide radicals occurs at the disk electrode while the oxidation of the residual superoxide radicals (that have not been scavenged by the chalcone) occurs at the ring electrode.

Reaction 1: Reduction of molecular oxygen at disk electrode
Disk current    O_2_ + e^−^ ⇌ O_2_•^−^(1)

Reverse Reaction 2: Oxidation of superoxide radicals at the ring electrode
Ring current    O_2_•^−^ ⇌ O_2_ + e^−^(2)

Thus, the rate at which increasing concentrations of Butein or DHDM scavenged the generated superoxide radicals during the electrolytic reaction was determined by obtaining the percent value of the quotient of the ring current and the disk current at each concentration. These percent values were denoted as the *collection efficiency* of each chalcone at different concentrations. Using Microsoft Excel, collective efficiency values were plotted against the corresponding concentrations of Butein or DHDM to produce a graph illustrating the effect of increasing concentrations of Butein or DHDM on the scavenging of superoxide radicals in the electrolytic solution. Ultimately, the slope of the curves served as a quantitative measure of the antioxidant activity of Butein and DHDM.

### 2.4. Computational Experiments: Docking Chalcones into Catalytic Site of Falcipain-2

Calculations were performed using programs from Biovia (SanDiego, CA, USA). Density functional theory (DFT) code DMol3 was applied to calculate energy, geometry, and frequencies implemented in Materials Studio 7.0 [27]. We employed the double numerical polarized (DNP) basis set that included all the occupied atomic orbitals plus a second set of valence atomic orbitals, and polarized d-valence orbitals [28]; the correlation generalized gradient approximation (GGA) was applied including Becke exchange [29], plus BLYP-D correlation including Grimme dispersion when van der Waals interactions were involved [30]. All electrons were treated explicitly and the real space cutoff of 5 Å was imposed for numerical integration of the Hamiltonian matrix elements. The self-consistent field convergence criterion was set to the root mean square change in the electronic density to be less than 10^−6^ electron/Å^3^. The convergence criteria applied during geometry optimization were 2.72 × 10^−4^ eV for energy and 0.054 eV/Å for force. Calculations were performed with no solvent inclusion, and in water solvent, using the continuous model of Dmol^3^ [31]. Docking studies were performed with the CDOCKER package in Discovery Studio 2020 version.

Cysteine protease inhibitors of Falcipain-2 have been shown to cause direct impairment of the hemoglobin degradation pathway in malaria pathogenesis [21,22,23]. Thus, to test if compounds **1**–**4** could be competitive inhibitors of Falcipain-2, we performed docking experiments using Discovery Studio software. The crystal structure of Falcipain-2 was downloaded from the Protein Data Bank (PDB file: 3BPF) and automatically prepared using a protocol embedded in the software. The active site of Falcipain-2 was located and our crystal structure coordinates of DHDM and Butein, respectively, were loaded into the active site following a docking procedure.

Homobutein (**3**) and 5-Prenylbutein (**4**) coordinates were calculated using Discovery Studio. Multiple positions were tested to determine the most preferred position of the ligand in terms of energy that allows for effective and strong binding to the active site. All receptor-ligand interactions were observed and recorded for later analysis. Discovery Studio provides two parameters for evaluation of poses obtained after docking: (1) “Cdocker interaction energy” is the non-bonded interaction energy (composed of the van der Waals term and the electrostatic term) between the protein and the ligand related to the force field CHARMm without including any solvation energy term; (2) “Calculate Binding Energies” protocol improves the estimation of binding free energy in a protein-ligand complex by including the solvation effect using the CHARMm Generalized Born based implicit solvation models. The binding energy is calculated using the following equation: Energy (Binding) = Energy (Complex) − Energy (Ligand) − Energy (Receptor).

## 3. Results

### 3.1. X-ray Crystallography

The 3D structures of Butein (**1**) and DHDM (**2**) were determined via single X-ray diffraction using a Bruker Apex II CCD Diffractometer and both chalcones were shown to be relatively planar structures. Analysis of the 3D structures of both chalcones was performed to identify intramolecular and intermolecular interactions including the atomic arrangements, bond angles, and torsion angles. Table 1 summarizes the crystal structure data for DHDM and Butein.

#### 3.1.1. Structural Characterization of 2′,4′-Dihydroxy-3,4-dimethoxychalcone (DHDM)

The structural analysis of DHDM shown in Figure 2, crystal data details in Table 1, revealed a mostly planar molecule [largest torsion angle C7-C8-C9-O5 is 16.3(2)°] that engages in limited intramolecular and intermolecular hydrogen bonding, intermolecular ring stacking. Figure 3 shows the ring stacking that occurs between molecules of DHDM. Figure 4 illustrates the intramolecular O6-H11…O5 2.538 (1) Å and the one intermolecular O7-H13…O5 2.714 (1) Å H-bond occurring among molecules of DHDM. There is a methoxy methyl-H/π hydrogen bond with distance C16-H161…centroid phenyl plane 2.747(1) Å. The effect of weak CH/π interactions in the molecular assembly can be seen in Figure 4 and their role in crystal structure assembly is recognized [32,33,34].

#### 3.1.2. Structural Characterization of Butein

Table 1 summarizes the crystal structure data for Butein monohydrate, and the results seen in Figure 5, shows the asymmetric unit to contain one Butein molecule and one water molecule. Figure 6 shows the hydrogen bonds that occur within and between molecules of Butein. Table 2 includes more detailed geometrical information on the hydrogen bonds of Butein. The shifted intermolecular ring stacking between Butein molecules is about 3.55 Å. The structure of Butein is similar to an earlier low resolution (R = 17%) structure determination [35].

Symmetry transformations used to generate equivalent atoms:#1*x* − 1, *y*, *z*#2*x* − 1, *y*, *z* + 1#3−*x* + 1, −*y* + 1, −*z* + 1#4−*x* + 1, −*y* + 1, −*z+2*#5*x* + 3/2, −*y* + 1/2, *z* − 1/2

### 3.2. Computational Experiments: Antioxidant Activity Studied Using DFT and Docking Chalcone into Catalytic Site of Falcipain-2

#### 3.2.1. DFT Studies

##### Butein

Butein X-ray coordinates were input for Materials Studio program DMOL^3^, and Figure 7 shows the relevant distances in the B ring of Butein; they will be of interest for comparison with the structural changes after scavenging the superoxide radical. Figure 7, Figure 8, Figure 9 and Figure 10 and Appendix A show the geometrical structures of the investigated species after DFT optimizations. Figure 8 shows the scavenging of superoxide by Butein. The initial separation between O atoms in the superoxide radical (1.373 Å) adjusts to 1.425 Å, meanwhile the 4-H(hydroxyl) separates from its 4-O(hydroxyl), 1.597 Å, due to its capture by superoxide, O-H distance of 1.027 Å. Thus, the superoxide determines the formation of a Butein radical species. In addition, if a proton is available, the O_2_H species will evolve towards H_2_O_2_, Appendix A; H_2_O_2_ formation has been demonstrated by us for other antioxidants [36,37]. When comparing Figure 9 with Figure 8 bond distances, the insertion of the unpaired electron induces ring B orbital localization, i.e., the newly formed C=O bond establishes double bond conjugation. Thus, the pair of C-C bonds (1.373 and 1.382 Å) are much shorter that the adjacent C-C bonds of 1.448 Å, 1,484 Å (next to the C4=O carbonyl) and those distant from the C=O carbonyl, (1.447 Å, 1.422 Å). Additional calculation for the semiquinone Butein, π approached by superoxide, indicates further scavenging (Figure 10). The original O–O separation in the superoxide becomes much shorter, 1.299 Å, forming a molecule of O_2_ linked to the polyphenol. The original separation between both radicals, 3.50 Å, becomes shorter, 2.948 Å. The new structural Butein species thus becomes anionic, due to capture of the superoxide electron. We conclude that Butein is able to scavenge at least 2 superoxide radicals. A conformational analysis was performed varying the torsion angle involving both aromatic groups. Figure 7 displays the DFT minimum energy molecule, the torsion angle involving the catechol ring, on the left of Figure 7, is −3.4°, slightly lower than −6.2° in the crystal. This angle was rotated every 15°, reaching 90°, and the corresponding single point energy calculated, Appendix A shows one these structures. Table 3 shows this variation in energy: the wider the torsion angle the greater the energy needed to reach the corresponding conformation. A similar calculation was performed varying the other aromatic ring torsion angle (Appendix A), which is −1.0° (minimum DFT) and −5.0° (crystal) and Table 3 also includes the corresponding energies. It is observed that the catechol torsion angle is energetically easier to modify than the non-catechol torsion angle (ring A) with the latter requiring approximately twice the energy, For instance, when the torsion angle is 60°, 6.0 kcal/mol are needed by the catechol ring B, and 13.0 kcal/mol for ring A. This is probably due to the intramolecular H bond between the 2′ hydroxyl and the chalcone carbonyl, which establishes an approximately planar conformation, that is more energetically demanding to twist than the planarity induced by the catechol ring.

The crystal structure of Butein contains a molecule of water and estimating its role in modifying the single molecule of Butein in the packing is not obvious, also because all 4 Butein hydroxyl groups establish important H-bonds useful for crystal and molecular interactions. A comparison between the packed crystalline molecule of Butein and the isolated DFT minimum structure is done by studying the molecular coplanarity of the two rings. Appendix A show the torsion angles associated with the minimum DFT energy: they are −1.0° (ring A), 178.4° (carbonyl), 179.1° (ethelyn) and −3.4° (catechol ring). In the crystal these 4 angles are −5.0°, −5.6°, 178.9° and −6.2°, respectively. It is clear that the energies involved in the packed crystal affects Butein coplanarity. However, a good agreement between the crystal and the DFT calculation can be concluded. In addition, from Table 3 the energy involved in torsion angle variation within 15°, corresponding to Butein coplanarity in the crystal, is very small.

##### DHDM

The scavenging of superoxide by DHDM is less efficient than that of Butein, Figure 11, Figure 12, Figure 13 and Figure 14. A comparison between the energies of DHDM-superoxide radical (Figure 12) and DHDM chalcone radical—O_2_H species after abstraction of the hydroxyl H (Figure 13), shows that the one in Figure 12 is 1.6 kcal/mol lower. In contrast with Butein (Figure 8), Figure 12 shows that for DHDM, the H is not captured by superoxide, O(chalcone)-H distance 1.037 Å (it was 1.530 Å in Figure 8).

However, as the RRDE experiment (shown below) indicates scavenging of superoxide by DHDM, we may conclude that the scavenging of superoxide by DHDM is due to the 1:1 DHDM:superoxide complex shown in Figure 12. This is suggested by the fact that the other potentially reactive H2′(hydroxy) is engaged in the intramolecular H-bond to O(carbonyl), and so less available for scavenging (result not shown). In addition, DHDM is a weaker scavenger of superoxide from the RRDE experiment, vide infra.

However, an alternative mode of scavenging the superoxide radical by DHDM was explored by placing the radical on top of ring A, 3.50 Å apart, to perform geometry minimization. This was chosen because the carbonyl located nearby this ring can have an electrophilic influence and so help to make the superoxide electron better suited for inclusion in the ring. On the contrary, ring B will have an opposite effect, due to the electron withdrawing effect of two methoxy groups. Figure 14 shows the minimum geometry obtained. There is shortening of the distance between the two centroids, 3.296 Å, compared with initial 3.50 Å, and variation of the associated bond O–O length of superoxide. Thus, the original 1.373 Å separating the two oxygen atoms in superoxide became 1.337 Å, while all distances in the ring are similar to those before the arrival of superoxide, shown in Figure 11. Therefore, a π–π complex between DHDM and the superoxide is stabilized. In this complex the separation between the two interacting moieties is less marked when compared to Butein, Figure 10 (2.948 Å). As seen in Figure 10, two radicals are interacting, and pairing their unpaired electrons is an important driving force.

##### Homobutein

The molecule of Butein was modified using an option in DMOL^3^ and the resulting geometry of Homobutein, having a methoxy group in position 3 instead of the hydroxyl, was geometrically minimized with DFT and shown in Figure 15. Then, H4(Homobutein) and O(superoxide) were placed at van der Waals separation (2.60 Å) and the geometry optimization shows O(superoxide)–H4 interaction of 1.581 Å, Figure 16 left. As the potential product (formation of O_2_H plus Homobutein semiquinone) has lower energy than that the reactant, a transition state (TS) was submitted, resulting in the feasible scavenging of superoxide, with ΔG of −1.9 kcal/mol and E(barrier) of 2.1 kcal/mol, Figure 16 and Figure 17. The TS was characterized by a unique imaginary frequency.

##### 5-Prenylbutein

5-Prenylbutein molecular structure was calculated as done earlier for Homobutein and DFT calculations were performed to obtain the minimum of energy. The interaction between 5-Prenylbutein and superoxide does not follow a similar trend as Butein. That is, the catechol moiety of the 5-Prenylbutein scavenger is not able to release its H4(hydroxyl) to the superoxide radical. Appendix A shows the approach of superoxide, and Appendix A the potential formation of O_2_H and 5-Prenylbutein semiquinone. As the former has 3.4 kcal/mol lower energy than the structure in Appendix A, we conclude that superoxide can only establish a complex such as indicated in Appendix A. An additional calculation was performed after placing [H_3_O]^+^ at van der Waals separation from the O_2_H moiety of Appendix A. The resulting geometry minimization generated a H_2_O_2_ molecule and the semiquinone 5-Prenylbutein radical, Appendix A. Thus, Butein is a better scavenger of superoxide than 5-Prenylbutein, as the release of H(4) is spontaneous in the former, while it needs a proton present in the latter. We conclude that Butein is a better scavenger for the superoxide radical than 5-Prenylbutein, DHDM and Homobutein.

#### 3.2.2. Docking of Butein and DHDM

X-ray coordinates of the 3D structures of compounds **1**, **2** were docked into the active site of 3BPF (Falcipain-2) [38]. This PDB file was downloaded and the CHARMm forcefield was applied. This includes assignment of H atoms. 3BPF contains the inhibitor E64, which was selected to generate the sphere of radius 15 Å, useful for docking, later the inhibitor was eliminated, and docking was performed.

The catalytic residues of falcipain-2 include an essential Cys42 amino acid that we hypothesize will chemically interact with the characteristic ethenyl-carbonyl group of the chalcone structures. A related bond formation was shown to be of importance in a recent study by our group where celastrol anti COVID-19 biological activity was correlated to covalent bond formation between Cys145 (active site of main protease SARS COVID-19) and a celastrol carbonyl [39].

##### Butein

Figure 18 displays relevant interactions for pose 1 of Butein at the active site of the falcipain-2 enzyme, obtained after docking into the 3BPF as retrieved from the PDB database. We see that S–H(Cys42) cleavage is suggested by two interactions: (a) H—O(carbonyl) distance of 2.066 Å, and (b) S(Cys42)–C(carbonyl) distance of 3.209 Å. H bonds between Butein ring B hydroxyls and Asp234 (2.194 and 2.055 Å) and π lone pair involving Butein ring A and Asn173 provide stability to the Butein-active site complex. Specific interactions are provided in Figure 19.

Discovery Studio molecular mechanics program [40] provides the CDocker interaction energy, which is the non-bonded interaction energy (composed of the van der Waals term and the electrostatic term) between the protein and the ligand related to the force field CHARMm. This item does not include solvent effects, and is −34.4 kcal/mol for Butein. When calculating binding energy (−17.7 kcal/mol), which includes solvent effects, for pose 1, the interaction between Cys42 and Butein is strengthened, with S–C(carbonyl) of 2.282 Å and O(carbonyl) binding H[S(Cys42)], that is confirming the docking results. Butein ring B hydroxyls H-bonds with Asp234 are also confirmed, but the π lone pair Butein involving ring A and Asn173 is lost, Figure 20. There were no clusters for these docked poses and analysis of poses 2–10 showed longer S(Cys42) interactions with the chalcone ethenyl carbonyl moiety, or no other potential covalent interactions.

##### DHDM

Most poses of DHDM (1,3-5,7,8,10) show no interaction between S(Cys42) and the ethenyl carbonyl moiety. The related docking analysis for DHDM chalcone is displayed in Figure 21 and Figure 22 for pose 2. CDocker interaction energy is −36.3 kcal/mol, only slightly more negative than that of Butein. From this docking, there is a weaker S(Cys42) interaction with C(carbonyl) than seen in Butein. However, S(Cys42) interaction with the ethenyl α carbon, 3.121 Å, suggests reactivity, assisted through Cys42 S-H cleavage that is induced by the O(carbonyl) capture of the H, 2.031 Å. After calculating binding energy, −21.1 kcal/mol, the structural pattern is confirmed with shortening of both S-C(α-ethenyl) distance, 2.794 Å, and O-H, 1.399 Å. This covalent DHDM Falcipain-2 complex is also stabilized by H-bonds, Ile85 and Asp234, and His174 π stacking amide with chalcone ring A, Appendix A. Poses 6 and 9 are closely related to pose 2 and suggest covalent interaction with Cys42, but when calculating binding energy pose 6 did not confirm expectations.

#### 3.2.3. Additional Docking on Known Antimalarial Chalcones

As it is of interest to compare the interaction of falcipain-2 active site with structurally related Butein chalcones in the literature, docking of Homobutein and 5-Prenylbutein, already demonstrated to be biologically active [13,14], was also performed.

##### Homobutein

Docked poses 2,3,6,7 form a cluster showing no covalent approach of S(thiolate) to the ethenyl-carbonyl moiety of Homobutein. Pose 6 and 7 include an H-bond to Leu172, which is missing in poses 2 and 3. After calculating bonding energy of these 4 poses, pose 6 shows the best score, −16.7 kcal/mol. In pose 6 the shortest separation is between S(Cys42) and the ethenyl β carbon, 3.559 Å (docking) and 3.400 Å (binding energy). Poses 4,5,8,9 also form a cluster, with no interactions between S(Cys42) and the ethenyl carbonyl moiety. Thus, there is no covalent interaction between Homobutein and the FP2 active site and we conclude that Homobutein can at most establish reversible inhibition. Figure 23 shows the H bound to the ethenyl β carbon hindering the approach of S(Cys42) to the chalcone β carbon and so a potential Michael addition seems not feasible. Figure 24 shows additional amino acid interaction to docked pose 6. Homobutein CDocker interaction energy is −31.2 kcal/mol for pose 6.

##### 5-Prenylbutein

In this study, Cys42 S-H cleavage is suggested by two interactions in pose 3: (a) short H–O(carbonyl) distance of 1.945 Å, and (b) S(Cys42)–C(carbonyl) distance of 3.728 Å, the latter being weaker than that seen in Butein, Figure 25. More importantly, the breaking of the intramolecular H bond between O(carbonyl) and 2′H(hydroxyl), a strong interaction always found in 2′-hydroxy chalcone crystal structures, suggests formation of a S-C(carbonyl) bond as a driving force. In addition, (a) an H bond between 4-H(hydroxyl) of 5-Prenylbutein and Asp234, and (b) π stacking between 5-Prenylbutein ring A and amide interaction with His174 and Asn173 (Figure 25) confer important stability to the 5-Prenylbutein falcipain-2 complex. The CDocker interaction energy for 5-Prenylbutein is −35.6 kcal/mol for depicted pose 3, and additional interactions are seen in Appendix A. Pose 3 is closely related to pose 4 and has some similarity with pose 10. These 3 poses form a cluster that involves the α,β-unsaturated ketone moiety bridging the two aromatic rings. The C=C double bond moiety is also exposed to potential nucleophilic attack from S(Cys25). Figure 26 shows this pattern for pose 3 where separation between S(Cys42) and the C(ethenyl) β carbon is 3.233 Å (3.648 Å, to the α carbon). Thus, a potential Michael reaction is suggested, where H[S(Cys42)] is captured by the chalcone O(carbonyl), to form an hydroxyl moiety. Meanwhile there is nucleophilic attack of S(Cys42) thiolate to the β C(ethenyl) atom and breaking the β carbon CH bond. In pose 10 the β C(ethenyl) atom is turned away from S(Cys42) and becomes unavailable for S(thiolate) nucleophilic attack, Figure 27. However, the α carbon is now closer to S(Cys42), 3.130 Å, in comparison with pose 3 and pose 4, (it was 3.648 Å in pose 3 and 3.912 Å in pose 4) and suggests nucleophilic attack from S(Cys42), Figure 27. We calculated the binding energy for poses 3 and 10, and they are markedly different, −30.8 and −17,.3 kcal/mol, respectively, which favors pose 3, Figure 28. The minimized obtained structure of pose 3 shows an O(Carbonyl)–H[S(Cys42)] short distance, 0.7 Å (a real O-H bond distance is about 1 Å), and S-C(ethenyl) β carbon of short distance, 2.40 Å. For pose 10, Figure 27, after calculating binding energy there is no possible covalent formation since no interaction between S(Cys42) and any potential C atom is seen. We conclude that pose 3 may follow a typical Michael addition, in agreement with the biological activity shown by 5-Prenylbutein [14]. Pose 3 has the best score in this series of chalcones interacting with FP-2: −30.8 kcal/mol, compared with DHDM (−21.1 kcal/mol), Butein (−17.7 kcal/mol) and Homobutein (−16.7 kcal/mol).

### 3.3. RRDE Antioxidant Activity Assay of 2′,4′-Dihydroxy-3,4-dimethoxychalcone (DHDM) and Butein

The effects of increasing concentrations of DHDM and Butein on the superoxide redox reaction are demonstrated through current density against potential curves plotted in Figure 29 and Figure 30 respectively.

In Figure 29 and Figure 30, the ring current is represented by the upper curves, while the disk current is represented by the lower curves. The generation of superoxide radicals from dioxygen occurs through a reduction reaction in the disk electrode process while the conversion of the superoxide radicals back to dioxygen (reverse reaction) occurs through oxidation in the ring electrode process. As shown by the voltammogram, this redox reaction is nearly completely reversible as, although the curves are initially separated from each other in the ring portion, the curves generated from the negative scanning of superoxide radicals overlap almost entirely with the curves generated from the positive scanning of oxygen.

For the RRDE experiments in this project, the disk potential was swept from 0.2 V to −1.2 V which was negative enough to generate superoxide radicals. Then, while keeping the ring potential at −0.25 V, the disk potential was swept back to 0.2 V which was positive enough to oxidize the superoxide radicals to dioxygen.

The collection efficiencies at increasing concentrations of DHDM and Butein were graphed (Figure 31 and Figure 32 respectively). Figure 31 demonstrated that as the concentration of DHDM increased in the electrolytic solution, there was a decrease in the collection efficiency, represented by the linear relationship (R^2^ = 0.9886) between both variables. Thus, the rate at which superoxide radicals decreased in the period between its production at the disk and its subsequent oxidation at the ring increased as more DHDM was added to the electrolytic solution, suggesting that DHDM was contributing to the scavenging of the superoxide radicals. Figure 32 demonstrated the same trend with scavenging of superoxide radicals by increasing concentrations of Butein occurring at a more rapid rate than DHDM, as depicted by the higher slope of the curve (−11.2 × 10^4^ for Butein compared to −8.0 × 10^4^ for DHDM). The antioxidant capability of both chalcones for scavenging the superoxide radical, using the same protocol, is in between quercetin (−15.4 × 10^4^) and embelin (−6.0 × 10^4^) [37].

Overall, the RRDE technique provided a quantitative measurement of scavenging superoxide radicals by DHDM and Butein (collection efficiency) with Butein being the relatively more effective antioxidant compound.

## 4. Discussion

In the fields of medicinal chemistry and pharmaceutical sciences, chalcones are widely known and used as templates for the synthesis of drugs [12]. The major classes of falcipain-2 inhibitors are peptides, closely followed by isoquinolines, thiosemicarbazones, and chalcones [23]. Hence, chalcones **1**–**4** are of interest and could serve as falcipain-2 inhibitors.

Comparing the results of X-ray analyses in Figure 4 and Figure 6, we see that Butein participates in a greater number of intermolecular interactions than DHDM due to its additional two hydroxyl functional groups. Hydrogen bonds are known to enhance ligand/drug-receptor interactions and enable high-affinity binding of lead compounds [41]. As Butein can participate in more hydrogen bonding than DHDM, it suggests that Butein has a greater capacity to interact with cellular receptors. 

From our computational analyses, the calculated 3D docking images reveal a closer proximity of S(Cys-42) to C(carbonyl) in the Butein-3BPF complex compared to that in the DHDM-3BPF. Thus, there is a higher probability of the formation of S(thiolate), after S-H cleavage, which can then establish an S-C(carbonyl) covalent bond between Butein and Cys42 in the active site of falcipain-2 than between DHDM and Cys42 (Figure 18 vs. Figure 20, respectively). Additionally, unlike in the DHDM-3BPF complex, the H(Cys-42) in the Butein-3BPF complex is in the right orientation and close distance to form a hydrogen bond with the O(carbonyl), facilitating S–H cleavage, which generates S(thiolate) useful for nucleophilic attack to the chalcone. After calculating binding energy bond distances of interest become shorter for both chalcones. However, Butein shows stronger covalent bond formation with S(Cys42).

Homobutein shows a weaker interaction with FP-2 active site as no indication of covalent bonds emerges from docking and there is also a weak O(carbonyl) contact with H[S(Cys42)]. The shortest separation for S(Cys42) is established with the ethenyl β carbon, suggesting a potential Michael addition, but its long distance, 3.400 Å, seems difficult to decrease, probably due to the H bound to the ethenyl β carbon. In fact, the expected direction for S(thiolate) attack to the nucleophilic target (approximately forming 90° with the C-H moiety) seems precluded as the S–C(β-carbon)–H angle is much smaller, 25.7°. Thus, the lack of covalent bond formation, suggests that at most Homobutein may be a reversible inhibitor of FP2.

On the other hand, it is of interest that our computational results show that the strong intramolecular H bond between O(carbonyl) and H(2′-hydroxyl) that is always present in 2′OH-chalcones related crystal structures [15], appears broken for the biologically active 5-Prenylbutein, when interacting with the falcipain-2 active site, Figure 26. This is a strong indicator of potential covalent bond formation between S(Cys42) and the C(carbonyl) and, interestingly, Butein also has strong associated indicators envisaging its potential related covalent affinity at the active site though C(carbonyl).

The scoring parameter in this molecular mechanics study suggests the decreasing order of inhibition following 5-Prenylbutein (−30.8 kcal/mol), DHDM (−21.1 kcal/mol), Butein (−17.7 kcal/mol), and Homobutein (−16.7 kcal/mol). On the other hand, the stronger S(thiolate)–nucleophile receptor interaction follows the order Butein [2.282 Å from C(carbonyl)], 5-Prenylbutein (2.410 Å from ethenyl β carbon), DHDM (2.794 Å from ethenyl α carbon), and Homobutein (3.400 Å from ethenyl β carbon), However, the scavenging of superoxide shows an important difference between Butein and 5-Prenylbutein. According to our DFT calculations, only having a proton in the biological environment of 5-Prenylbutein will allow to scavenge the radical using the catechol moiety, whereas Butein shows spontaneous scavenging.

Previous studies have shown that the number and location of hydroxyl groups can affect the scavenging ability of antioxidants [26]. Specifically, an increase in the number of hydroxyl groups at the ortho position in a phenol compound denotes an increase in antioxidant activity [42]. Hence, phenol compounds with two hydroxyl groups at the ortho position (catechols), have been suggested to be the most active antioxidants [43]. Thus, the catechol nature of Butein coupled with its two additional hydroxyl groups may explain its higher collection efficiency and better antioxidant capacity.

Therefore, we conclude that Butein can serve as an excellent inhibitor as it can establish the shortest covalent bond with the Cys42 active site of 3BPF. Since the CDocker interaction energy indicator of stability between the docked compounds **1**–**4** and the active site of falcipain-2 are similar, (ranging from −32.9 to −36.3 kcal/mol), no conclusions can be inferred from this potential discriminatory parameter. 

The collection efficiency vs. concentration curve generated from the RRDE voltammetry experiment of Butein (Figure 32) has the same characteristic feature (negative slope) as shown by quercetin [37]. Comparison of slopes in Figure 31 and Figure 32 indicates that Butein is a much stronger antioxidant than DHDM as it can more efficiently clear the toxic superoxide radicals generated by *Plasmodium* parasites in the host cells. This is also supported theoretically by our DFT results, showing Butein to be more effective scavenger than DHDM. In fact, Butein is able to sequester easily at least 2 superoxide radicals, Figure 8 and Figure 10, whereas, in contrast, DHDM only captures one, Figure 14. When comparing Butein and its homologue catechol derivative 5-Prenylbutein, some loss of antioxidant activity is seen in the latter, although the presence of an acidic environment helps it to scavenge superoxide. Homobutein is an intermediate case between Butein and DHDM, as it can sequester superoxide with its 4-OH by overcoming a mild barrier (2.1 kcal/mol), Figure 16 and Figure 17, which makes Homobutein a good scavenger.

## 5. Conclusions

Overall, analyses of the X-ray structures, as well as the results of computational studies (docking and DFT) and antioxidant RRDE voltammetry experiments, show that Butein (**1**) may serve as an effective antimalarial drug molecule due to (a) its greater intermolecular interactions, (b) its predicted inhibitory effect on falcipain-2, and (c) its higher antioxidant activity.

Molecular mechanics (docking and minimization of binding energy) at the active site of falcipain-2 show in this study the same property for all four chalcones: O(carbonyl) is able to capture H[S(Cys42)] and generate S(Cys42) thiolate, useful for further nucleophilic attack to the chalcones. Only Homobutein has this weak interaction.

Docking results of all four chalcones placed at the active site of falcipain-2, show that 5-Prenylbutein (**4**) has a unique property: the ability to break the strong intramolecular 2′-H(hydroxyl)–O(carbonyl) H bond present in 2′-hydroxychalcone crystal structures. However, 5-Prenylbutein shows a different covalent interaction (obtained through a typical Michael addition) than Butein, with a longer and weaker S(Cys42)–C(ethenyl) β carbon distance of 2.410 Å, compared to Butein S(Cys42)–C(carbonyl) bond, 2.282 Å. Hence, (**4**) possesses a weaker specific binding feature (not including additional amino acid interactions) and so Butein appears better suited for establishing a covalent complex at the active site of falcipain-2. DHDM (**2**) and Homobutein (**3**) show even weaker interactions with S(Cys42) thiolate. Thus, for DHDM there is reactivity of S(thiolate) towards the ethelyn α carbon (S-C distance of 2.794 Å), which is longer than that seen in (**1**) and (**2**). Finally, Homobutein S(Cys42) thiolate has a weak interaction with C(ethenyl) α carbon, 3.400 Å, but this is even weaker than for the other three chalcones.

In summary, this study shows a large versatility of possible chalcone binding sites when interacting with the falcipain-2 active site Cys42 amino acid: the C(carbonyl) of Butein; the C(ethelyn) β carbon of 5-Prenylbutein; the C(ethelyn) α carbon for DHDM and a very weak C(ethelyn) β carbon for Homobutein.

This set of four 2′,4′-di-hydroxy chalcones, has diverse substitution in ring B (Figure 1). Thus, for Butein there are two hydroxyls in position 3, 4; for Homobutein one hydroxyl in position 4 and one methoxy group in position 3; for DHDM two methoxy groups in position 3,4, and shows a marked trend in antioxidant activity. Butein is the best scavenger of superoxide (capturing 2 radicals, using features located in ring B), followed by Homobutein, which captures only one superoxide on ring B. In contrast, DHDM, not having hydroxyls in ring B, uses an alternative π–π mode of scavenging, using ring A features, and which may be also feasible in compounds (**1**) and (**2**). Thus, the increasing number of hydroxyls correlates with scavenging activity [26], but the particular chalcone structure (two aromatic rings A and B that are joined by a three-carbon α,β-unsaturated ketone system) facilitates scavenging making chalcones noteworthy compounds. It is also interesting that the catechol moiety of 5-Prenylbutein, also existing in Butein, is markedly weakened for scavenging superoxide. In fact, a proton presence in the environment is needed to release the H-4 atom of 5-Prenylbutein, in contrast with Butein that shows no barrier for H-4 release.

Inhibition of falcipain-2 would prevent or reduce hemoglobin degradation and thereby slow down the progress of the malaria infection. In addition, all of these chalcones can chelate iron cations in heme that would otherwise be consumed by *Plasmodium* parasites and enhance their replication [17,20]. Butein as an antioxidant catechol compound can also contribute to better scavenging of superoxide radicals that are toxic to the host cells and utilized by *Plasmodium* parasites [44] as shown here by RRDE and DFT results. In conclusion, our findings provide more evidence for the use of chalcones **1**–**4** as starting points for the development of novel, effective, and affordable antimalarial drugs.

## Figures and Tables

**Figure 1 molecules-26-06511-f001:**
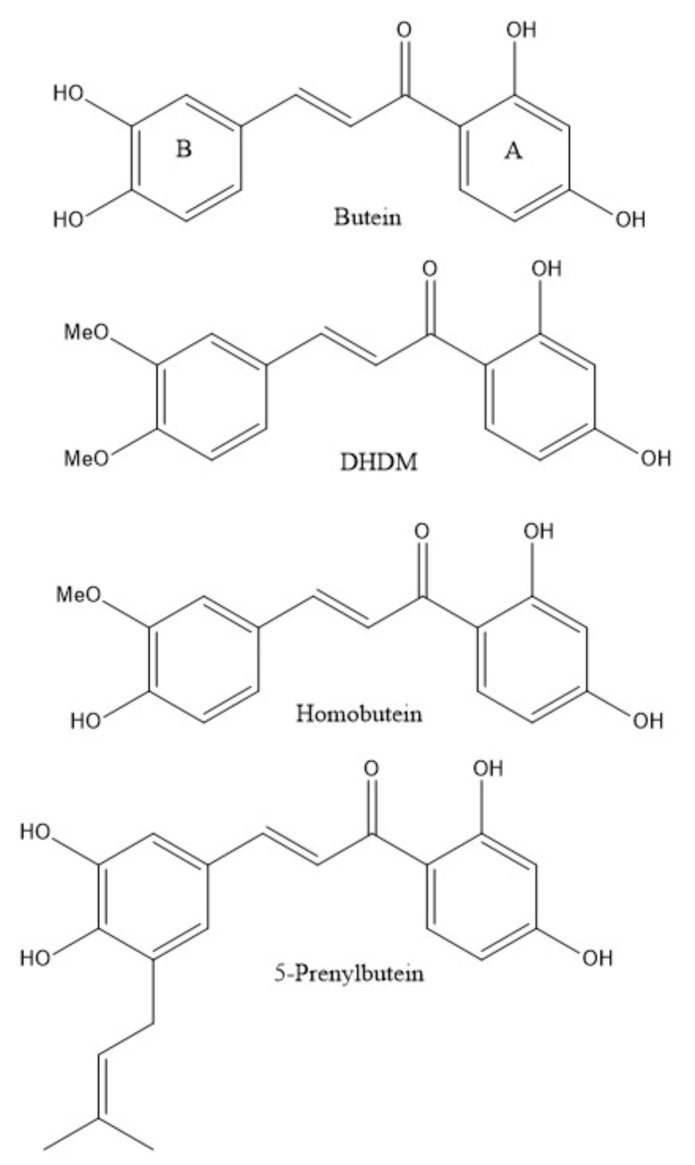
The 2D chemical structures of studied 2′,4′-diOH chalcones, (**1**) Butein, (**2**) DHDM, (**3**) Homobutein, and (**4**) 5-Prenylbutein. The substituent variation in these compounds is only in ring B, shown on the left of each chalcone.

**Figure 2 molecules-26-06511-f002:**
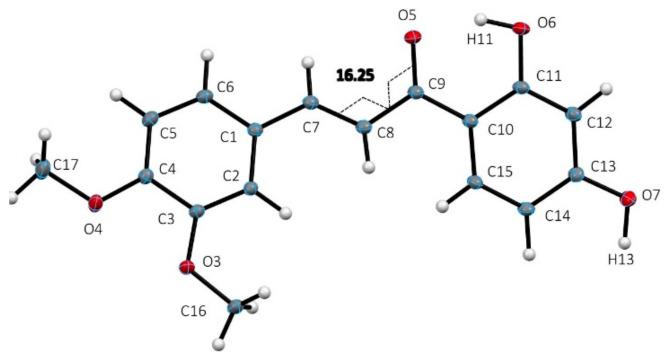
The 3D molecular structure of 2′,4′-dihydroxy-3,4-dimethoxychalcone (**2**, DHDM). Largest torsion angle C7-C8-C9-O5 is 16.3(2)°. Anisotropic displacement parameter ellipsoids are drawn at 50% probability.

**Figure 3 molecules-26-06511-f003:**
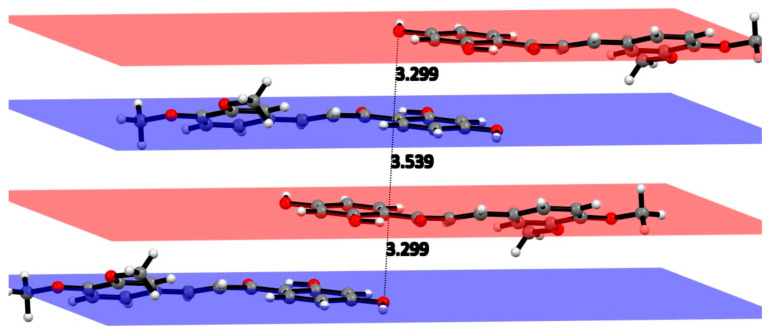
Shifted ring stacking interactions between molecules of DHDM. The intermolecular distances alternate between 3.299 and 3.539 Å.

**Figure 4 molecules-26-06511-f004:**
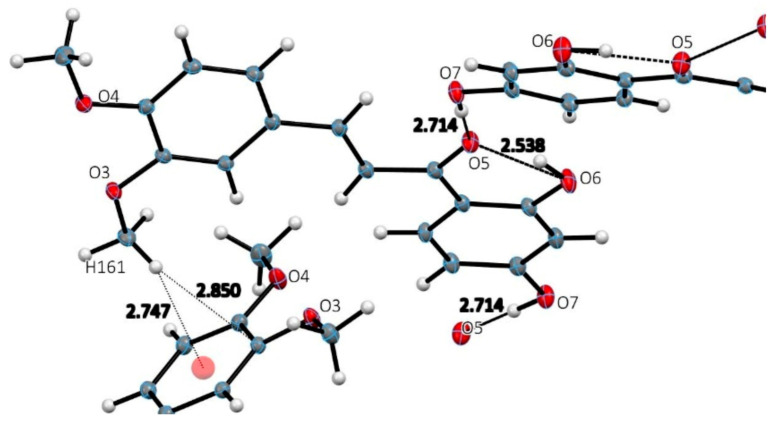
Hydrogen bonding interactions in DHDM crystal structure. One intramolecular O6-H11…O5 2.538 (1)Å and one intermolecular H-bond O7-H13…O5 2.714 (1) Å. There is a methoxy H-π hydrogen bond C16-H161…phenyl centroid 2.747 (1) Å; (methoxy C16-H161…C3 2.850 (1) Å).

**Figure 5 molecules-26-06511-f005:**
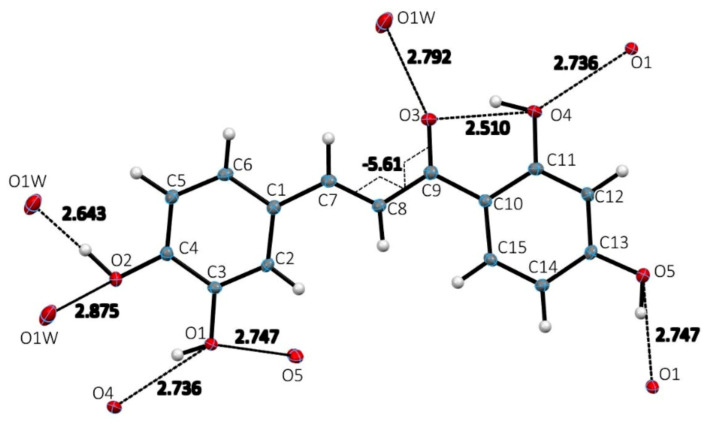
Atomic labeling, hydrogen bond distances surrounding single molecule and torsion angle in the molecular structure of Butein monohydrate. Three of four OH-groups, O2-H4, O1-H3, O4-H11 have two H-bonds. Hydroxyl O5-H13 has only one H-bond. Torsion angle C7-C8-C9-O3 is −5.61°. Ellipsoids represent 50% anisotropic displacement parameters.

**Figure 6 molecules-26-06511-f006:**
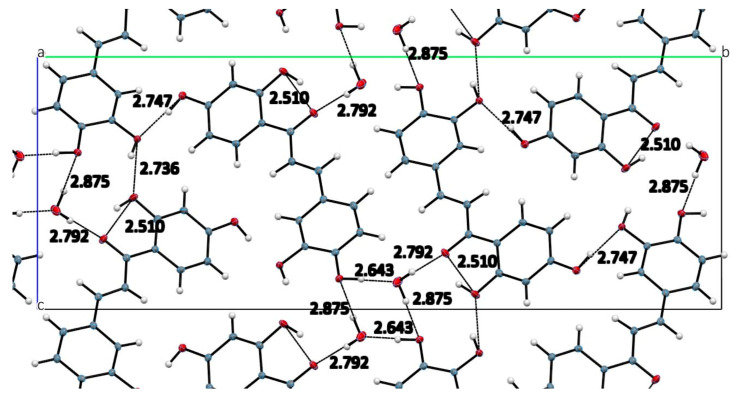
Repeating H-bond interactions including those with water molecule viewed down *a* axis. Geometrical details are in Table 2.

**Figure 7 molecules-26-06511-f007:**
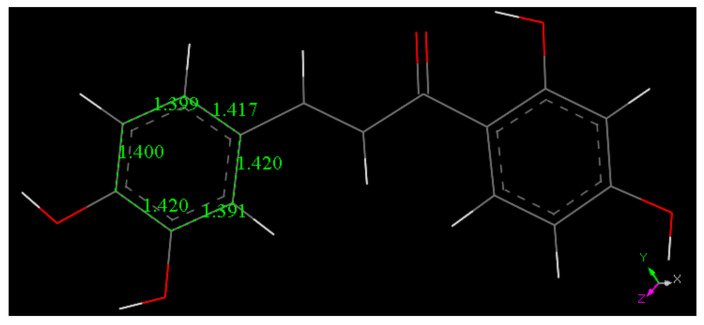
DFT minimum of energy structure of Butein (**1**). Relevant bond distances of ring B are indicated.

**Figure 8 molecules-26-06511-f008:**
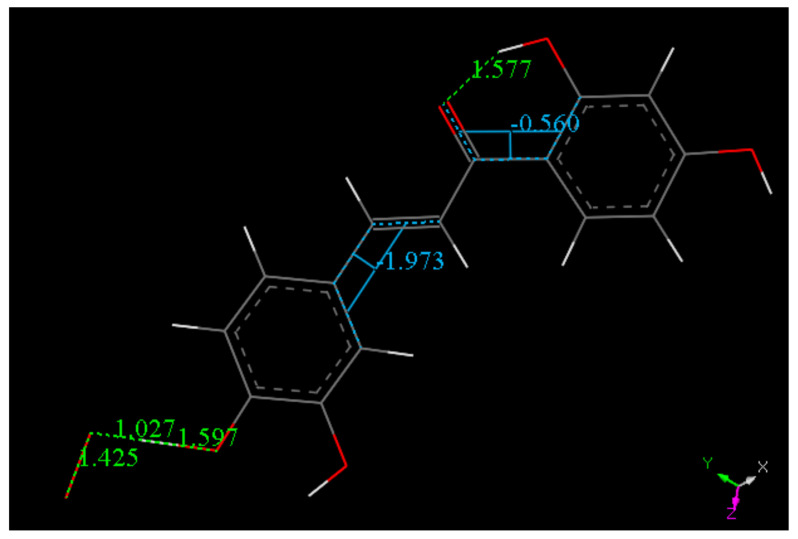
Converged DFT minimum of energy arrangement after geometry optimization for van der Waals separated (2.60 Å) superoxide and 4-H(hydroxyl) of Butein. Calculation includes water solvent and dispersion term Grimme. It is seen that H(4) is captured by superoxide (O–H distance 1.027 Å).

**Figure 9 molecules-26-06511-f009:**
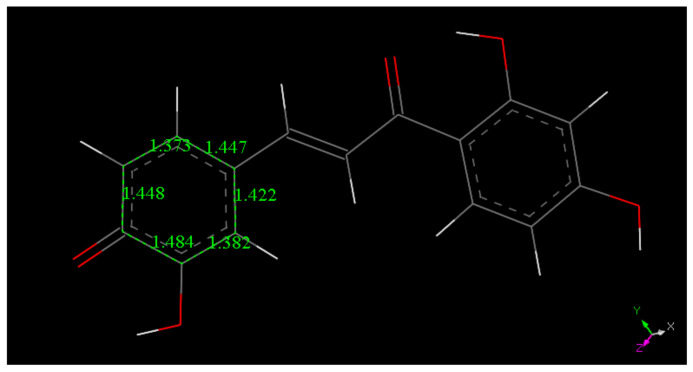
After elimination of O_2_H from Figure 8, the semiquinone Butein radical structure is geometry optimized. Comparison with Figure 7 shows aromatization loss due to the double bond character of the newly formed carbonyl.

**Figure 10 molecules-26-06511-f010:**
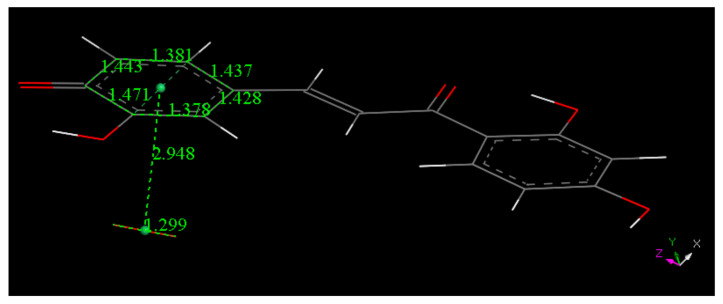
The Butein semiquinone radical of Figure 9 reacts with an additional superoxide radical, π–π approach, to produce this minimum energy structure after geometry optimization. The interacted ring B is flat, due to the O(carbonyl)–H(hydroxyl) hydrogen bond, and so the alternative attack of superoxide from above the ring should produce an equivalent minimum.

**Figure 11 molecules-26-06511-f011:**
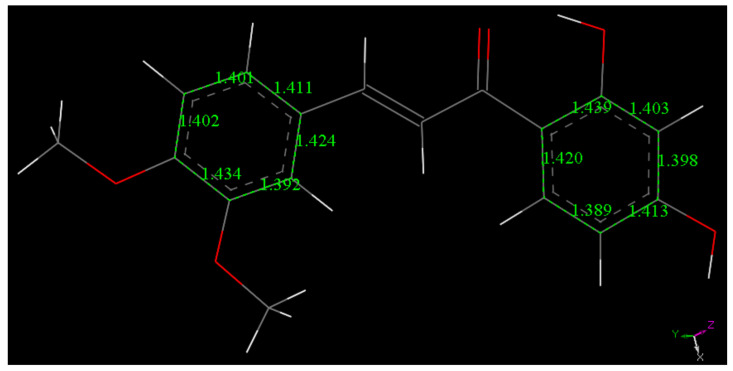
The chalcone (**2**), DHDM, minimum energy structure obtained after geometry minimization from initial X-ray coordinates.

**Figure 12 molecules-26-06511-f012:**
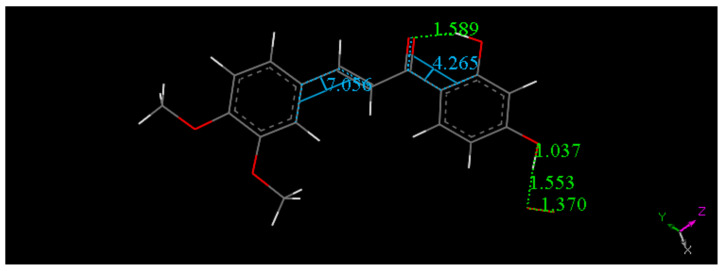
Resulting geometry minimization for H(hydroxyl) in position 4′of DHDM chalcone, approached by superoxide (initially separated by van der Waals distance 2.6 Å). Distances shown in this picture are minimized in water solvent including dispersion term Grimme. The 4′-hydroxyl H atom is not captured by superoxide, O(superoxide)- H distance is 1.553 Å.

**Figure 13 molecules-26-06511-f013:**
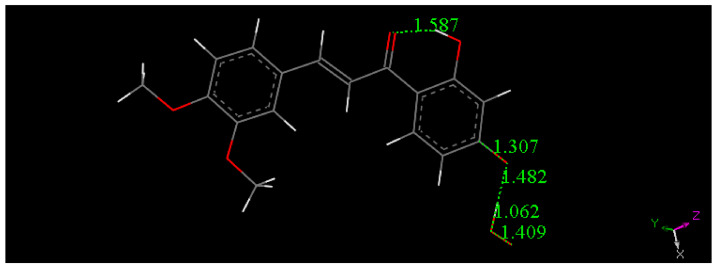
Result of geometry optimization after potential H(hydroxyl) capture by van der Waals separated superoxide to form O_2_H, bottom right, and DHDM semiquinone. This calculation shows no capture of H by the semiquinone, O-H distance of 1.482 Å. Calculation is done using water solvent and dispersion term Grimme. The Figure 12 structure is 1.6 kcal/mol lower in energy.

**Figure 14 molecules-26-06511-f014:**
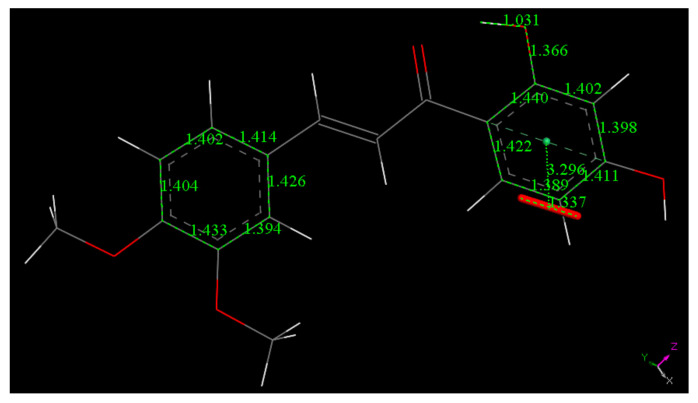
Result of DFT minimization for a π–π approach of superoxide on stacking DHDM chalcone ring A. The initial separation between both centroids was 3.50 Å. This is shortened to 3.296 Å, while the initial bond distance of O atoms in superoxide was 1.373 Å, shortened to 1.337 Å. Ring C–C bond lengths are not modified when this radical complex is formed, as seen comparing related data in Figure 11.

**Figure 15 molecules-26-06511-f015:**
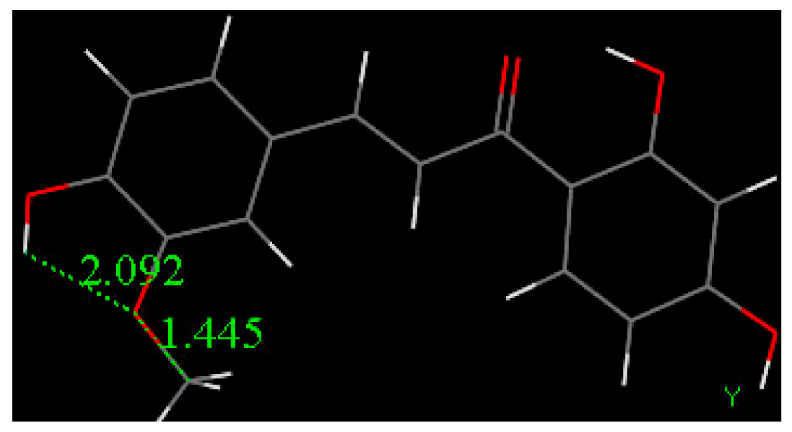
DFT geometry optimization of an isolated molecule of Homobutein, shows an intramolecular H-bond between H4(hydroxy) and O-3(methoxy), 2.092 Å.

**Figure 16 molecules-26-06511-f016:**
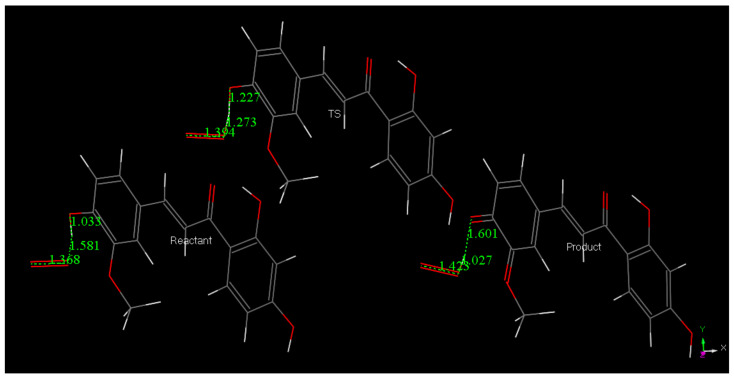
The DFT reactivity between H(4)Homobutein and superoxide shows capture of H4(hydroxy) by the radical, characterized by a transition state, ΔG of −1.9 kcal/mol, E(barrier) of 2.1 kcal/mol.

**Figure 17 molecules-26-06511-f017:**
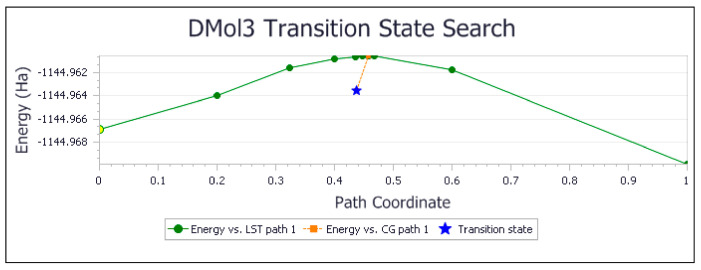
Energy profile for the transition state search between H(4)Homobutein and superoxide.

**Figure 18 molecules-26-06511-f018:**
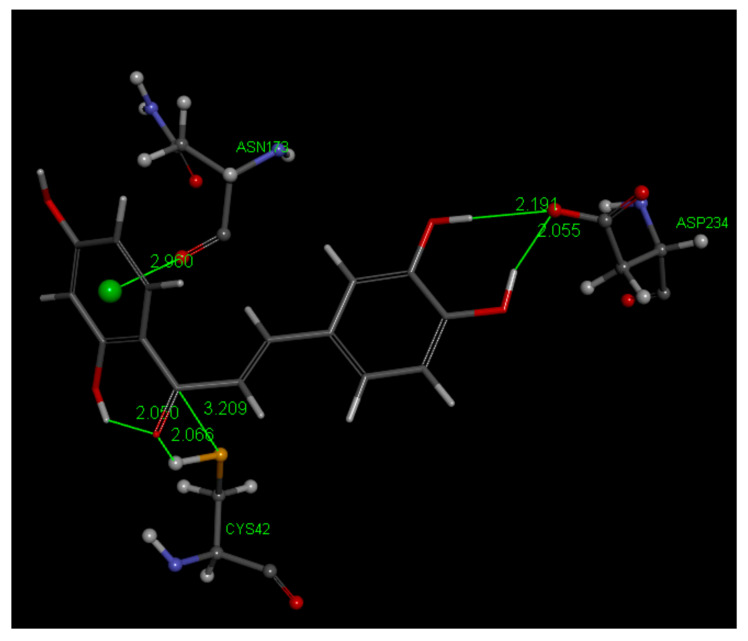
Pose 1 after docking Butein (**1**) in the 3BPF receptor.

**Figure 19 molecules-26-06511-f019:**
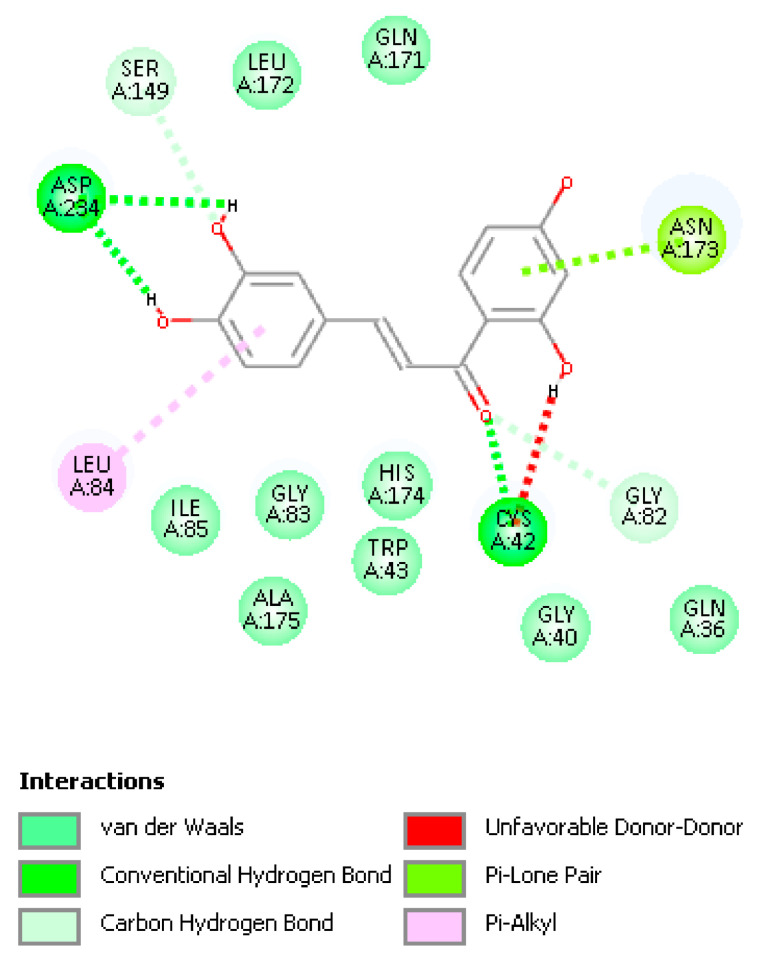
2D interactions between pose 1 Butein and the active site of falcipain-2. Unfavored Butein H(hydroxyl) in position 2′ suggests a potential break of the O(carbonyl)–H2′ H bond.

**Figure 20 molecules-26-06511-f020:**
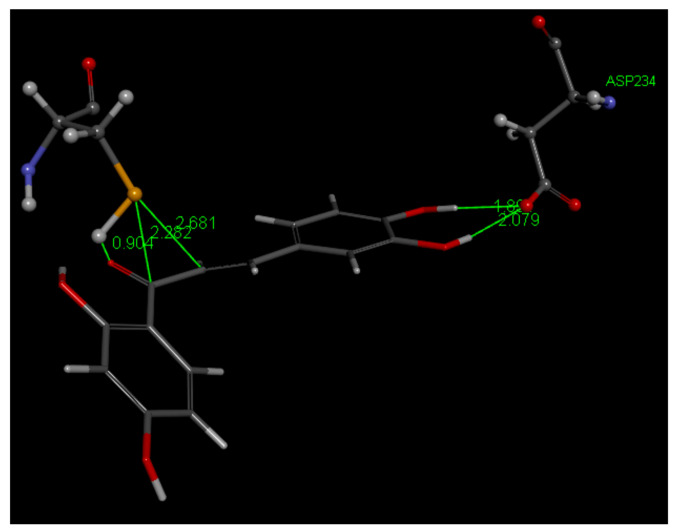
Butein (**1**) in the 3BPF receptor, pose 1, after calculating binding energies, −17.7 kcal/mol. H[S(Cys42)] is captured by chalcone O(carbonyl). S(thiolate) is 2.282 Å from C(carbonyl), suggesting a covalent bond, whereas the ethenyl α carbon is more distant, 2.681 Å, and seems not suited for S(thiolate) nucleophicic attack. Adding stability to the Butein falcipain complex there are H bonds from the catechol moiety of ring B to Asp244, and chalcone O(carbonyl) to polypeptide HN(Trp43), omitted for clarity.

**Figure 21 molecules-26-06511-f021:**
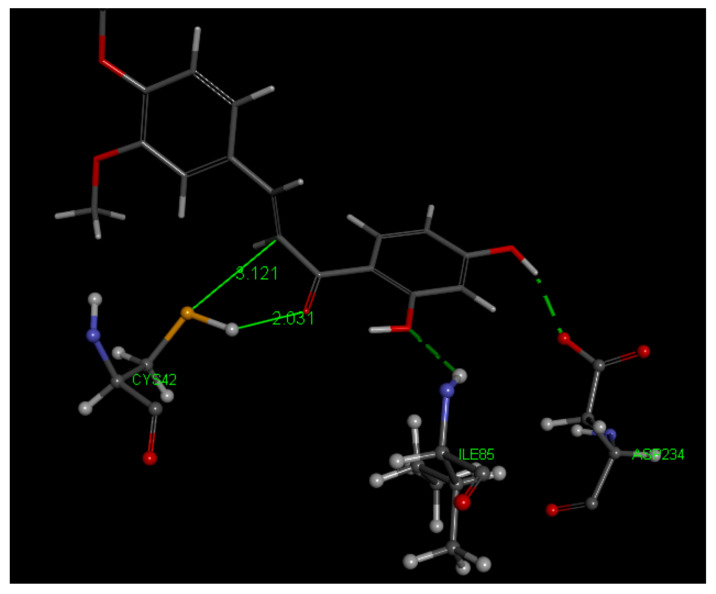
Pose 2 after docking DHDM (**2**) in the 3BPF receptor. The closest interaction for S(Cys42) to the chalcone involves the ethenyl α carbon, 3.121 Å. H bonds with Ile85 and Asp234 are also indicated.

**Figure 22 molecules-26-06511-f022:**
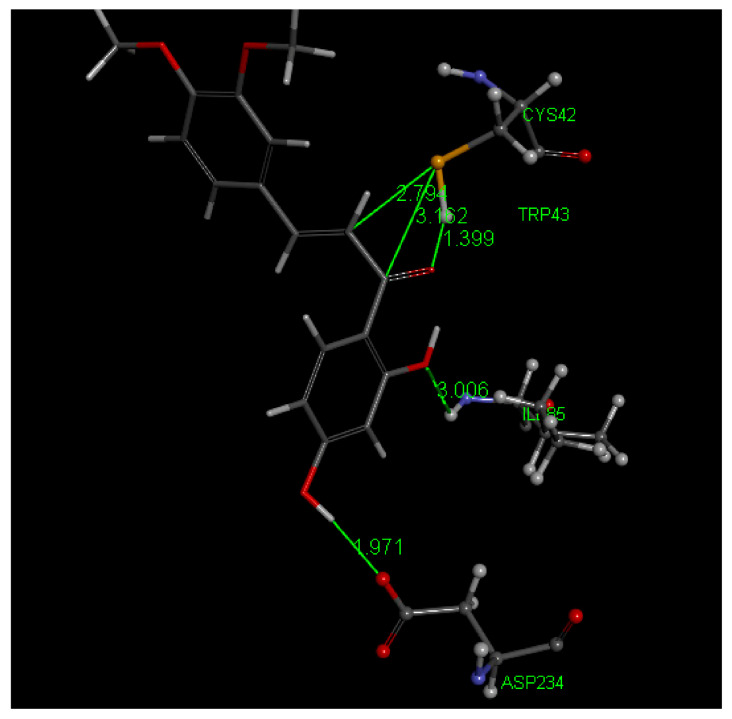
DHDM pose 2 from docking after calculate binding energy. The potential reactivity of S(thiolate) towards the ethenyl α carbon is confirmed by S–C distance of 2.794 Å, and shortening of H bond between H[(Cys42)] and O(carbonyl), 1.399 Å. H bonds involving chalcone and the two amino acids, Asp234 and Ile85, are also confirmed. Omitted for clarity is a H bond between Trp43 and O(carbonyl) of DHDM.

**Figure 23 molecules-26-06511-f023:**
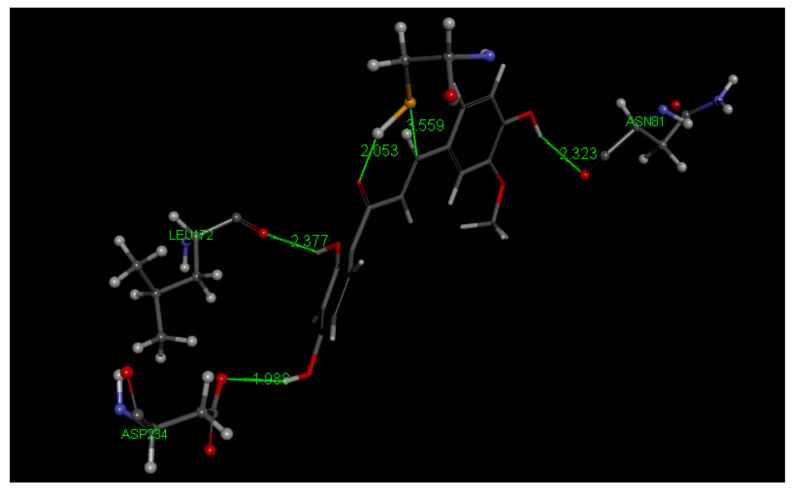
Three-dimensional (3D) interaction of Homobutein (**3**) with the receptor 3BPF Falcipain-2, pose 6. Compared to Butein, S(Cys42) shows a greater separation to the chalcone, 3.559 Å to the ethenyl β carbon. Three amino acids give stability to this arrangement through H bonds. O(carbonyl) shows a H-bond to H[S(Cys42)], 2.053 Å. Upon calculating binding energy (−16.7 kcal/mol) the arrangement is confirmed, showing S—C(β ethelnyl) separation of 3.400 Å, while O(carbonyl)—H[S(Cys42)] distance becomes longer, 2.458 Å. After minimization H bonds between Homobutein and amino acids Asn81, Leu172 and Asp234 show shorter lengths, 2.183 Å, 2.344 Å and 1.792 Å, respectively. The H atom bound to the ethenyl β carbon (shown in ball style) seems to hinder the approach of S(Cys42) and so no covalent bond seems feasible for Homobutein.

**Figure 24 molecules-26-06511-f024:**
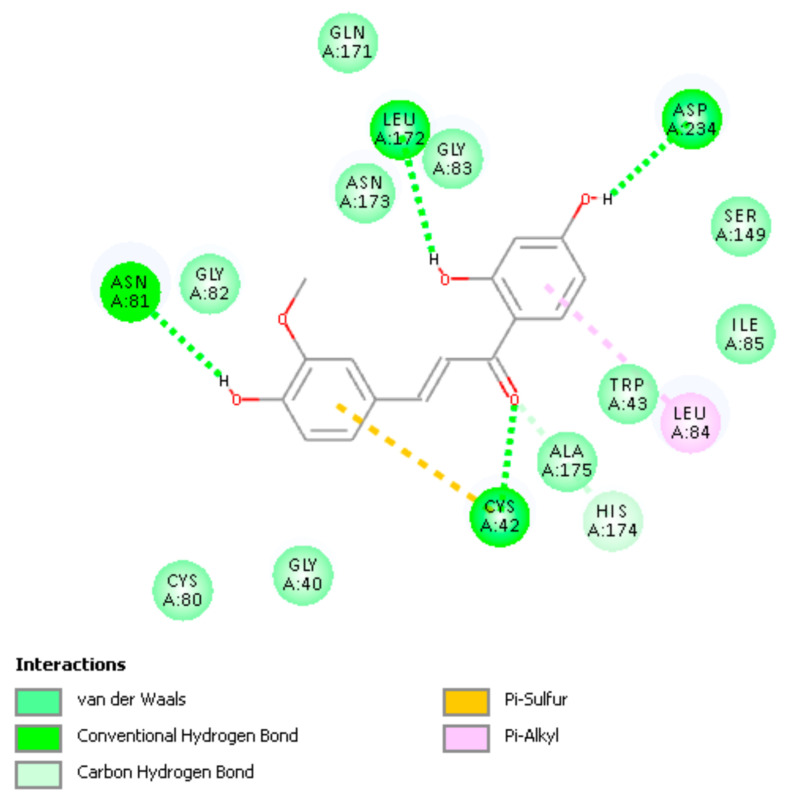
Two-dimensional (2D) pose 6 interactions between Homobutein and the active site of Falcipain-2.

**Figure 25 molecules-26-06511-f025:**
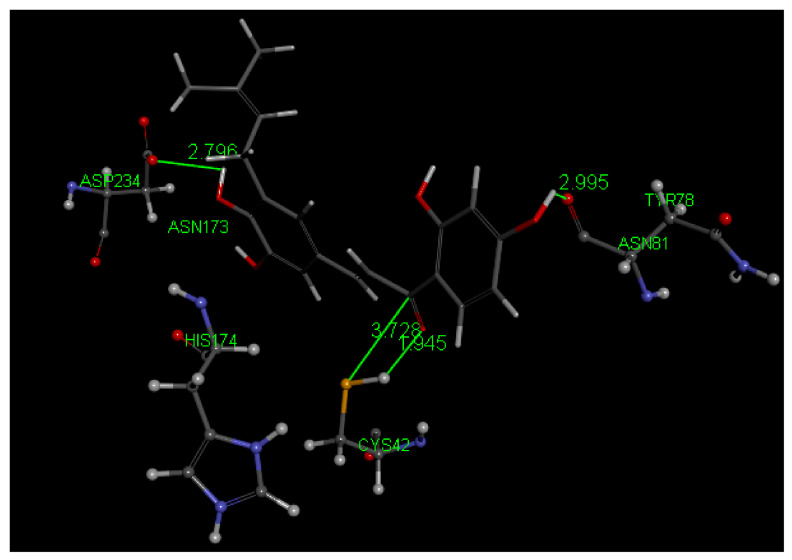
Pose 3, after docking 5-Prenylbutein (**4**) in the 3BPF receptor.

**Figure 26 molecules-26-06511-f026:**
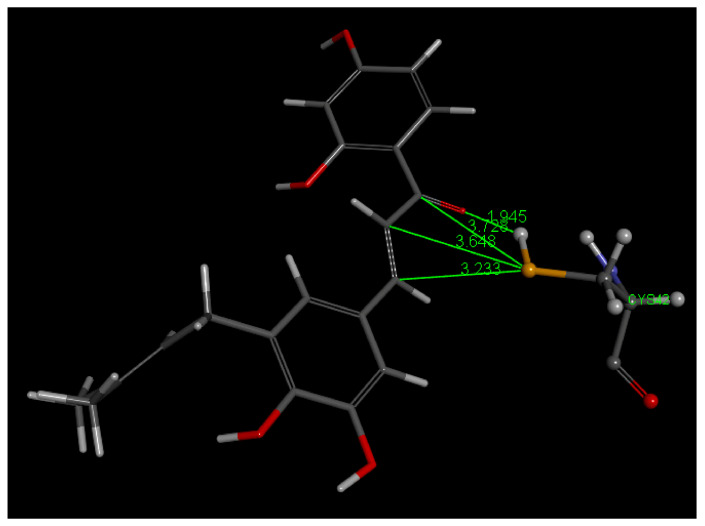
This is pose 3 after docking 5-Prenylbutein (**4**) in the 3BPF receptor, showing more details. A potential capture of H[S(Cys42)] by chalcone O(carbonyl), 1.945 Å, is envisioned, along with S(thiolate) nucleophilic attack to C(carbonyl), 3.728 Å, or, more probably, C(ethenyl) β carbon, 3.233 Å.

**Figure 27 molecules-26-06511-f027:**
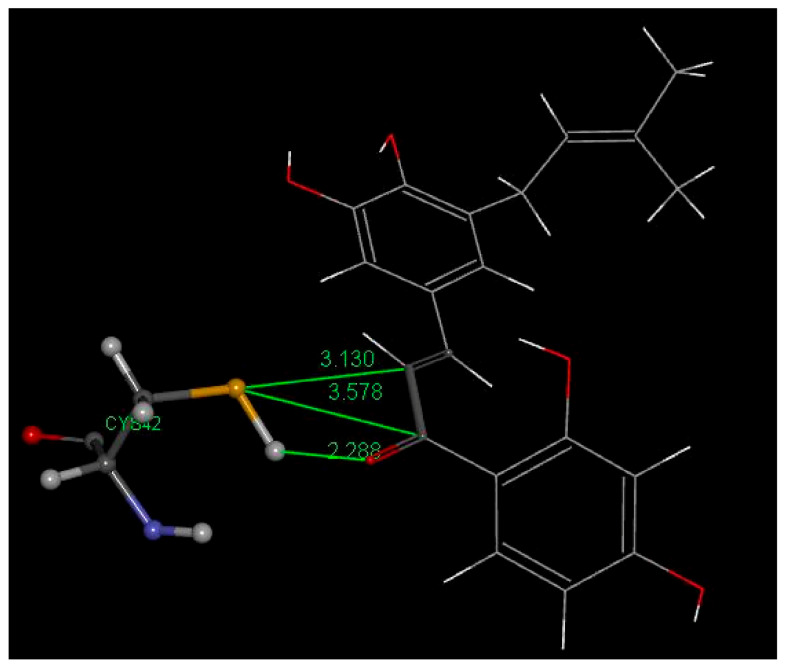
Pose 10, after docking 5-Prenylbutein (**4**) in the 3BPF receptor. A potential capture of H[S(Cys42)] by chalcone O(carbonyl), 2.228 Å, is envisioned, along with S(thiolate) nucleophilic attack to C(carbonyl), 3.578 Å, or, more probably C(ethenyl) α carbon, 3.130 Å.

**Figure 28 molecules-26-06511-f028:**
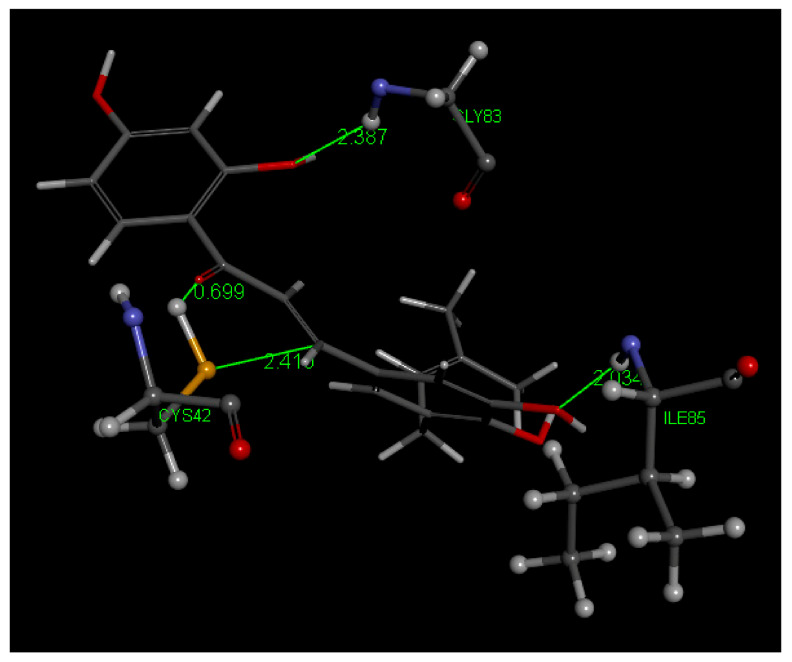
5-Prenylbutein (**4**) in the 3BPF receptor, pose 3, after calculating binding energies. S(Cys42) establishes a covalent bond with C(ethenyl) β carbon, 2.410 Å, and there is capture of H[S(Cys42)] by O(carbonyl). An additional interaction is established by Gly40, which makes a π amide stack, omitted for clarity.

**Figure 29 molecules-26-06511-f029:**
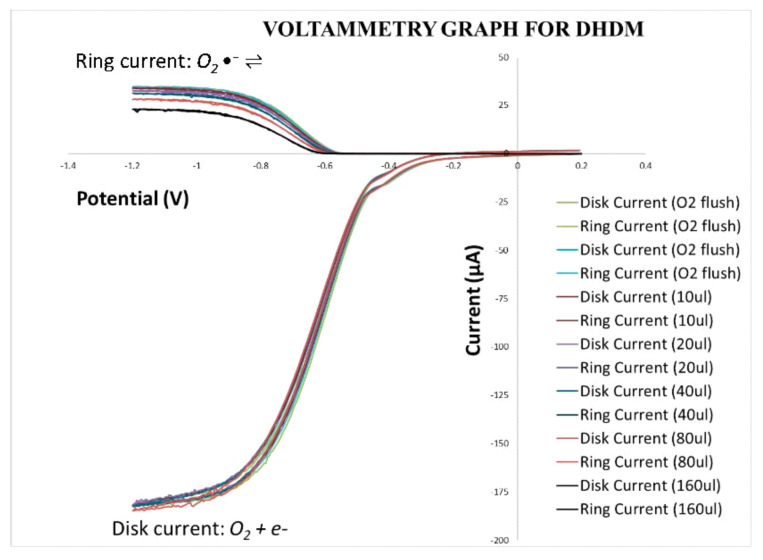
Effect of increasing concentrations of DHDM on current-density vs. potential curves.

**Figure 30 molecules-26-06511-f030:**
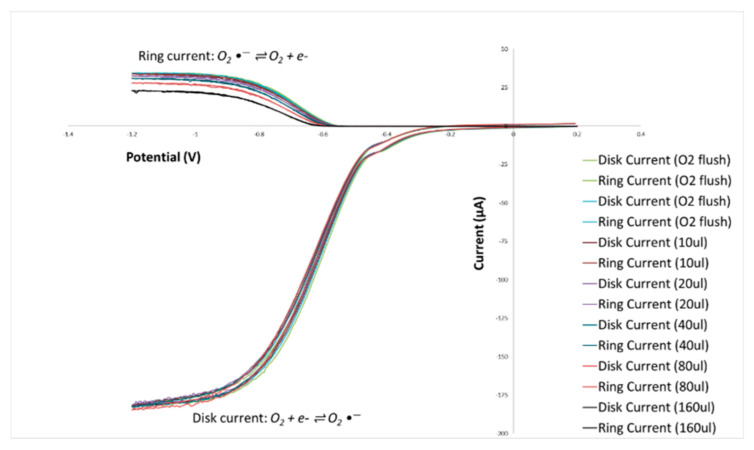
Effect of increasing concentrations of Butein on current-density vs. potential curves.

**Figure 31 molecules-26-06511-f031:**
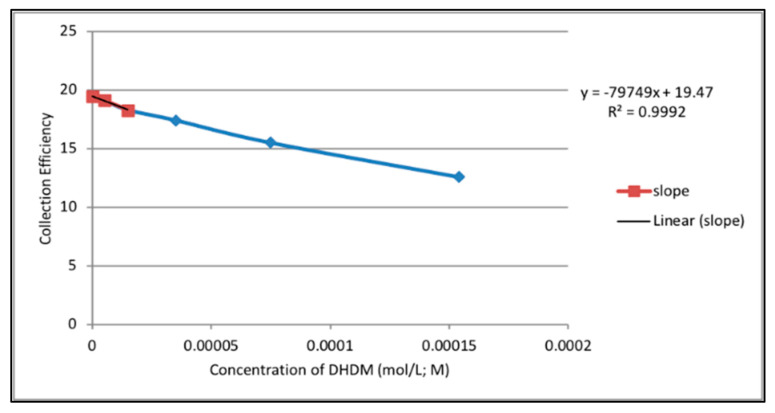
Collection efficiency for DHDM.

**Figure 32 molecules-26-06511-f032:**
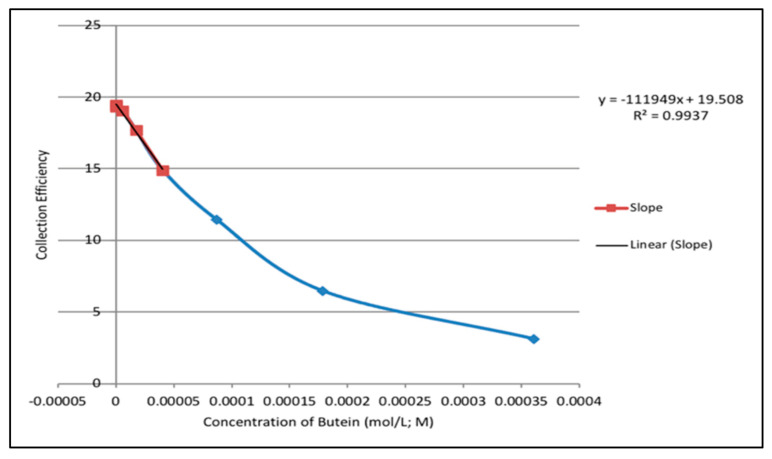
Collection efficiency for Butein.

**Table 1 molecules-26-06511-t001:** Crystal Data of 2′,4′-dihydroxy-3,4-dimethoxy chalcone, (DHDM) and Butein.

Chemical Compound	2′,4′-Dihydroxy-3,4-Dimethoxy Chalcone (DHDM)	Butein
Chemical formula	C_17_H_16_O_5_	C_15_H_14_O_6_
Formula weight	300.30 g/mol	290.26 g/mol
Temperature	125 K	125 K
Wavelength	0.71073 Å	0.71073 Å
Crystal size	0.050 × 0.080 × 0.310 mm	0.030 × 0.040 × 0.190 mm
Crystal system	orthorhombic	monoclinic
Crystal habit/color	clear yellow needle	clear yellow plate
Space group	*P b c a*	*P 2* _1_ */n*
Unit Cell Dimensions	*a* = 15.5081(11) Å	*a* = 4.2416(5) Å
	*b* = 7.9651(5) Å	*b* = 28.896(3) Å
	*c* = 23.7534(16) Å	*c* = 10.8045(13) Å
		*β* = 99.604(2)°
Volume	2934.1(3) Å^3^	1305.7(3) Å^3^
Z	8	4
Density (calculated)	1.360 g/cm^3^	1.477 g/cm^3^
Absorption coefficient	0.100 mm^−1^	0.115 mm^−1^
F (000)	1264	608
Diffractometer software	Bruker SAINT software	Bruker SAINT software
Absorption correction	Multi-Scan (SADABS)	Multi-Scan (SADABS)
No. of measured, independent reflections	70049, 4679	25876, 2883
Max, min transmission	0.9950 and 0.9700	0.9970 and 0.9780
R(int)	0.0467	0.1121
Final R indices, data with I > 2σ(I)	3581 data; R1 = 0.0461, wR2 = 0.1247	1603 data; I > 2σ(I)R1 = 0.0394, wR2 = 0.0676
Final R indices, All data	4679 data; R1 = 0.0652, wR2 = 0.1352	2883 data; R1 = 0.0954, wR2 = 0.0788
No. of parameters	263	246
No. of restraints	0	0

**Table 2 molecules-26-06511-t002:** Hydrogen bond distances (Å) and angles (°) for Butein crystal structure.

	Donor-H	Acceptor-H	Donor-Acceptor	Angle
O1W-H2W⋯O3#3	0.90(3)	1.90(3)	2.792(2)	174(3)
O4-H11⋯O3	0.96(2)	1.63(3)	2.5102(18)	151(2)
O5-H13⋯O1#5	0.91(2)	1.85(2)	2.7468(19)	168(2)
O1-H3⋯O4#2	0.83(2)	1.94(2)	2.7362(19)	158(2)
O1-H3⋯O2	0.83(2)	2.32(2)	2.7367(19)	111.2(1.8)
O2-H4⋯O1W#1	1.00(2)	1.64(2)	2.642(2)	176(2)
C5-H5⋯O1W#1	1.018(17)	2.636(17)	3.353(3)	127.4(1.2)
O1W-H1W⋯O2#4	0.92(3)	1.98(3)	2.875(2)	166(3)

**Table 3 molecules-26-06511-t003:** Conformational variation of Butein. Individual single point energy calculations were performed for the torsion angle associated with the catechol ring (central column) or to the non-catechol ring. They were subtracted by the energy of DFT minimum obtaining ΔEnergy. Torsion angles were fixed every 15°.

Catechol and Non-Catechol Fixed Torsion Angle (°)	ΔEnergy for Ring B Torsion Angle Variation (kcal/mol)	ΔEnergy for Ring A Torsion Angle Variation (kcal/mol)
15	0.04	1.14
30	1.48	4.83
45	3.61	9.17
60	6.02	13.05
75	7.80	15.81
90	8.47	16.85

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
