# Peer review of "X-ray Structure Determination, Antioxidant Voltammetry Studies of Butein and 2′,4′-Dihydroxy-3,4-dimethoxychalcone. Computational Studies of 4 Structurally Related 2′,4′-diOH Chalcones to Examine Their Antimalarial Activity by Binding to Falcipain-2"

_molecules, 2021, doi:10.3390/molecules26216511_

Round 1
Reviewer 1 Report
The paper presents a study on an interesting and timely topic including experimental and computational determinations. Though the employed techniques are adequate for the purpose of the study, the paper, particularly for the computational part, is written very poorly, often using a non-scientific language and the description of the computational procedure and some of the conclusions which are drawn result very questionable.
The division of Section 2 in subsections is very strange. For instance, 2.3. Computational Experiments: Docking Chalcones into Catalytic Site of Falcipain-2 and 2.4.3. DFT and Docking Calculations are given in different subsections and subsubsections and they should be in the same one. The whole section should be written with more care, it is difficult for the reader to go and find information in the paper.
Page 6, line 215 “BLYP correlation including Grimme code”. I think the authors mean Grimme’s correction. Is this BLYP-D in DMoL3 code? (By the way the name of the code is often written in different ways throughout the paper). A reference pointing to the kind of correction used should be inserted.
DFT studies: though the used procedure is not standard and rather questionable (the kinetics and dynamical behavior cannot be inferred from optimization calculations), the rationale of the study should be clearly explained and warnings on the kind of the information which can be obtained should be given. It should be stated from the beginning, for instance, that the investigated molecule and the superoxide radical are initially placed at an arbitrary interacting distance (which in principle should be larger) and then allowed to relax. The possibility of forming van der Waals complexes should also be explored.
The employed terminology should be fully revised. For instance, page 10, line 282: “Figures 7-13 show some DFT calculations.”?? In case, “Figures 7-13 show the geometrical structures of the investigated species after DFT optimizations”. This is just an example, the paper is full of such approximate statements.
Line 287: “if a proton is available, the O2H species will evolve towards H2O2, not shown”. Do the authors mean a hydrogen instead of a proton? Because otherwise a charged species should be formed. The charge should be preserved. “Not shown”: why? Is this just a hypothesis of the authors or the results of such calculations are not shown? This is a scientific paper, it is ok if it is just a conjecture (quite plausible by the way), but it should clearly emerge from the text.
Line 293: “Additional calculation for the semiquinone Butein, π approached by superoxide”, π approached should be better explained, besides there are two locations for the aromatic π electrons. Even worse, Caption of Figure 10 “Figure 10. The Butein semiquinone radical of Figure 9 reacts with an additional superoxide radical, π-π approach, to produce this minimum energy structure after geometry optimization.” There are no π-π interactions in the system. Besides the reaction is not shown in the figure (so the word “reacts” is not appropriate). Once more, these are minimum energy calculations which can tell about the thermodynamics and the geometrical structure of the minima, other data should be justified by additional calculations or described as hypothesis.
Line 361: “A comparison between the minimum of energy of Figures 12 and 13, shows that the one in Figure 12 is 1.6 kcal/mol lower”, I guess they mean “A comparison between the energies of the complexes DHDM chalcone-superoxide radical (Figure 12) and DHDM chalcone radical - O2H species after abstraction of the hydroxyl H (Figure 13)”. The following discussion in the paragraph does not have much physical sense: a difference in energy (suppose it is a ΔG) of 1.6 kcal/mol means an equilibrium constant at room temperature of 0.52, meaning that 1/3 of DHDM chalcone has reacted once the equilibrium has been reached. Even more so at physiological temperature. So the reaction would be thermodynamically feasible, the problem is whether it is also kinetically accessible. To verify that the transition state should be found, and the activation energy determined. This procedure should indeed be the proper one for all the investigated species: the thermodynamic might be favorable, but the kinetics might prevent the reaction to take place in a reasonable time (even when the reaction is exothermic, as in the butein case). I strongly suggest adding a study of the kinetics for all the investigated species.
Lines 388-390: “ The molecule of butein was modified using an option in DMOL3 and the resulting geometry minimized Homobutein is shown in Figure 14. Therefore, H4(Homobutein) and O(superoxide) were posed at van der Waals separation (2.60 Å)” Which option in DMoL3? What is the effect? “Therefore” should be “Then”. “posed” should be “placed”.
Docking Studies:
Lines 442-444: The procedure should be better described
Lines 459-462: It is not clear. As far as I can understand from the text the CDocker parameter (parameter?) is an estimate of the non-covalent binding energy resulting from the docking, therefore -34.4 kcal/mol is the estimated binding energy. I think it should be made clearer.
Other issues throughout the paper:
The number of significant figures in the distance values is different in different part of the text. This has to be corrected.
ΔE and ΔG were always written as Delta E and Delta G. This is not acceptable.
A comparison with crystallographic distances (possibly in a Table to be inserted in the Supplementary Material), besides that reported for torsional angle, should also be included to definitely assess the quality of the employed level of theory.
In short, in my opinion, the paper should be re-written and motivated in a much more careful fashion before it might be considered for publication in Molecules.
Reviewer 2 Report
General comments
The Manuscript ID molecules-1361607, entitled "X-Ray Structure Determination, Antioxidant Voltammetry Studies of Butein and 2’,4’-Dihydroxy-3,4-dimethoxychalcone. Computational Studies of 4 Structurally Related 2’,4’-diOH Chalcones to Examine their Antimalarial Activity by Binding to Falcipain-2" is study with significant topic and important goal for the use of chalcones 1-4 as starting points for the development of novel, effective, and affordable antimalarial drugs.
Specific comment
Title is too long and hard to follow. It should be shortened.
Abstract is too long and burdensome. It should be rephrased to be more informative.

Reviewer 3 Report
I’ve found this work very interesting. Studying Butein (which is an ethnopharmacological active compound) and three derivatives for their antioxidative properties and their potential Falcipain inhibition ability is sound, and important in the context of potent anti-malarial drug search.
These four compounds are not described equally, as only Butein and DHDM were studied by X-ray Crystallography. Is there a particular reason for that?
Overall, the paper clarity can be improved:
- Produce better pictures: figs 7-14 (and S1-S6) are hard to “read”. An easy fix would be to always use “stick” representation mode for superoxide and interacting groups just as it is done Fig. S7
- 14 is duplicated (but not in the text which refers to fig. 15)
- I suggest to use a proper atom labelling scheme to describe interactions, both in the DFT and docking studies results.
Now the annoying part:
A docking experiment usually produces several possible poses, ranked using a scoring function. The experiments related here are described on the basis of (I guess) the “best” pose, using a single scoring function which is not described (or even named) in the method part of the manuscript.
Hence, we are quite far here from an ideal “consensus scoring” approach…
For a more convincing conclusion related to the docking part, the authors should provide more details (at least in supplementary info). For instance, a superimposition of say the 5 or 10 best poses should be given associated to their respective scores, allowing the reader to know “how far” is the selected (interpreted!) pose compared to the other ones, that the authors have chosen not to consider.
I mean, what if Nb 2 pose of Butein contradicts the conclusions, with a CDocker parameter only 0.1 kcal/mol worse than the one for Butein?
Oppositely, the conclusions might be strengthened if other well-ranked poses present key interactions between Falcipain residues (especially Cys 42) similar to the ones described using the top ranked pose!
Another possibility could be to use more scoring functions (I doubt Discovery Studio provides only one scoring function, or a scoring function based on a single sets of force field parameter) in a rescoring approach.
Round 2
Reviewer 3 Report
OK. I think the paper can be published in this form.
This manuscript is a resubmission of an earlier submission. The following is a list of the peer review reports and author responses from that submission.
Round 1
Reviewer 1 Report
The paper presents a joined experimental – computational study on four butein-related chalcones showing antimalarial activity. The topic is surely of interest and timely, however the study is quite narrow in the number of investigated molecules, presenting only very limited variations and in the type of studies.
In particular, there are several points that the authors should address:
- the X-ray structure was only determined for the bare molecules under study and not for their complex with Falcipain-2, which is much more interesting and is indeed one of the aim of the paper, as highlighted by the docking studies. The complex between Falcipain-2 and at least one of the chalcones should be characterized.
- The DFT studies can indeed shed some light onto the ability of these molecules as superoxide scavengers. However, it is quite difficult to understand the point of the study before reading the end of the related paragraph: it should be better explained from the start. More importantly, i) the performance of the employed level of theory should be tested. For instance, by its capability of reproducing the crystal structure geometry upon optimization (here the crystal structure geometry is only taken as a starting point for optimization, no further checks were made); ii) It seems that only one of the hydroxyl groups in butein or in DHDM was examined when considering the interaction with superoxide ion. What guarantees that the examined one is that leading to the lowest interaction energy? All of them should be tested and a conformational study should be performed.; iii) the reaction path was not examined (i.e. possible transition state and separated products) nor, apparently, the charge distribution on the final products (OH2- and butein radical). How can the authors tell that it is not OH2 radical and butein anion?; iv) related to the preceding point, all the calculations were performed in gas phase, however the reaction is supposed to take place in biological medium and thus in solution (unless a hydrophobic pocket is considered). But all the calculations were carried out in gas phase. This is a crucial point, because water solution can affect the outcome of the reaction particularly when charged species are at play. v) A table with the interaction energies should be explicitly given. From what emerges from the text it seems that DHDM interacts with superoxide ions in stronger fashion. In short, the conclusions on the mechanism are only obtained on the basis of very fragile preliminary calculations. The reaction mechanism should be more deeply investigated.
- Docking studies: since there are no crystal structures of the complex between the investigated molecules and Falcipain-2, the crystal structure of Falcipain-2 was downloaded from PDB. i) Details on how the active site was located should be given. ii) More importantly, an explanation of why the authors elected Cys42 as the probable binding site should be given. iii) the authors write: “Cys42 S-H cleavage is suggested by two interactions” as docking study can only identify binding poses of non-covalently interacting partners. The possibility of the formation of a covalent compound is thus supposed on the basis of interaction distances, only giving a slight indication for the formation of a favorable complex. Therefore, conclusions on the possibility of the S-H cleavage cannot be drawn solely from such weak data. The interaction geometries (the most favorable pose) should be taken as a starting point for subsequent QM/MM calculations or QM (DFT) calculations on a smaller model showing the covalent reaction. A deeper study is needed to shed meaningful conclusion.
- The quality of the figures is very poor. Figure 1 keeps track of Words blank spaces, etc; Figs 7-21 are very difficult to read, they are too dark, too small and in some cases they are cut.
In my opinion, all these major revisions should be addresses before the paper is suitable for publication in Molecules.
Minor points: the Abstract is too long; A few misprints: for instance Figure DFT-1 instead of Figure 7.
Reviewer 2 Report
Review report for Manuscript ID: molecules-1299334
X-Ray Structure Determination, Antioxidant Voltammetry Studies of Butein and 2’,4’-Dihydroxy-3,4-dimethoxychalcone. Computational Studies of 4 Structurally Related Butein Chalcones to Examine their Antimalarial Activity by Binding to Falcipain-2 by Miriam Rossi et al. The authors have used chalcones butein, 2’,4’-dihydroxy-3,4-dimethoxychalcone (2, DHDM), Homobutein (3) and 5-Prenylbutein (4) to elucidate possible molecular mechanisms by which these compounds can provide therapeutic bebefits against malaria parasites. The use of X-ray diffraction, antioxidant activity by RRDE and computational study against falcipain-2 shows that butein is the most potent amongst the studied chalcones.
General comments-
In the abstract and Introduction it was stated that this paper is focussed on the study of Erythrina abyssinica, a tree native to Southern and Eastern Africa, and the Democratic Republic of Congo, for its constituents [specifically homobutein (3) and 5-prenylbutein (4) mentioned in paper] that are used to treat malaria in these regions. But in line 98 “Butein is a chalcone obtained from natural plants such as the heartwood of Dalbergia odorifera” which is the studied compound along with DHDM. Some revision to the introduction/abstract seems necessary to define the exact goals of this study. The exact nature for choosing butein and DHDM over the homobutein (3) and 5-prenylbutein (4) for the Xray and RRDE should be clarified.
My observation-Butein and DHDM are well suited to the study goals and they show the importance of free 3,4-dihydroxyl and dimethoxylated groups for antioxidant activity. While homobutein also would have two centers for superoxide interaction (the 4-hydroxy in both aromatic rings). In case of DHDM, since the catechol group was blocked the 4’-hydroxy from the acetophenone side of DHDM has shown interaction in the geometry optimization data. Additional geometry optimization data for homobutein and 5-prenyl butein should also be presented, since these are constituents of the Erythrina abyssinica tree. Overall the discussion will have to include the data of geometry optimization for homobutein and 5-prenyl butein for effectively deciding which chalcones is theoretically better suited for antimalarial compound.
In light of the above comments, the title also will need revision in the present situation, since only data of two butein derivatives was presented for X-Ray Structure Determination, and Antioxidant Voltammetry, and 4 derivatives computational studies.
Please refer to text in “Antiplasmodial natural products: an update by Nasir Tajuddeen and Fanie R. Van Heer, Malar J (2019) 18:404; https://doi.org/10.1186/s12936-019-3026-1
Page 34/62 in the above reference has some data on prenylated flavonoids and the reference cited therein may be helpful to present your results on 5-prenyl butein with more clarity.
Abstract-
Statement in line 17-20 to be revised- “Here, we describe results from experimental and computational investigations of 4 structurally related chalcones, Butein (1), 2’,4’-dihydroxy-3,4-dimethoxychalcone (2, DHDM), Homobutein (3) and 5-Prenylbutein (4) to elucidate possible molecular mechanisms by which these compounds clear malaria parasites.”
No in-vivo or in-vitro studies have been done to claim the above statement. Based only computational studies, XRD and RRDE can you be assured that these chalcones can clear the malaria infection?
Include common names in parenthesis for lines 76-77 “Azadirachta indica, Neem Tree, and Moringa oleifera, Drumstick Tree”
Lines 90-92 “Homobutein (3) and 5-Prenylbutein (4) are chalcones, frequently produced as a part of the plant’s defense system protecting against exposure to ultraviolet rays, pathogens, and toxins through their anti-inflammatory and high antioxidant activity” need proper citations dealing with reports on their anti-inflammatory and anti-oxidant activity. Current self-citation is insufficient.
Table 1 needs proper formatting into three independent columns. Some data of table 2 also needs formatting as footnote below the table 2. (line 258 onwards on page 11)
Only the most important crystal structure which defines the object of the present study should be retained in the manuscript along with the data presented in tables. The remaining data can be presented in the supplementary file, with a clear indication in the main manuscript wherever required.
Same goes for the computational data, only the most important to be presented in the main manuscript, remaining to be shifted to supplementary file with due citation.
Insert DOI for all citations in the references section.
Round 2
Reviewer 1 Report
I am afraid I do not find any improvement upon revision. The authors only addressed very few of my concerns, particularly the most technical ones which are basic to test the soundness of the work.
Indeed, though I can understand that the characterization of the complex between Falcipain-2 and the chalcones might be a complicate task and could be avoided by more seriously motivating the choice of the binding site (which is not done), all the computational issues meant to solve the flaws or eliminate any doubt on the physical meaning of calculations have not been taken into account.
In particular, no test with a DFT functional including corrections for dispersion forces has been used or at least tested: BLYP (the authors state they wrote PBE in the first version of the manuscript as a mistake, which is somehow sign of a careless attitude, on technical information which is fundamental to understand the reliability of the results) does not provide a good treatment of van der Waals interactions, whereas Grimme’s D-BYLP, for instance, does (but there are many others: cam-B3LYP, D-B3LYP). At least a test on one of the molecules to understand the impact on such calculations must be performed.
No comparison with the crystal structure of the chalcones molecules and the calculated ones was performed. In the authors’ response they just say “The optimized geometry of the chalcone skeleton for Butein and DHDM correspond very closely to that experimentally observed in the crystal structure” and nothing is added in the text. A comparison between the experimental structure when available and the calculated one is mandatory to test the reliability of the employed level of theory and is the standard in such kind of investigations. Furthermore, just stating that they match very closely without providing any table with actual numbers (even in the Supporting Information) is very questionable. If the authors feel that some bias can arise because intermolecular interactions are present in the crystal structures, dimers could be considered for the calculations.
No preliminary conformational study to assess the structure used in the subsequent calculations was performed.
Calculations in water solvent: the methods employed (a continuum solvent model, or a Poisson-Boltzmann approach or other) should be clearly specified in the computational details section.
The issues concerning the computational part of the work should be solved before I can recommend the paper for publication.
Reviewer 2 Report
Review report for Manuscript ID: molecules-1299334v2
X-Ray Structure Determination, Antioxidant Voltammetry Studies of Butein and 2’,4’-Dihydroxy-3,4-dimethoxychalcone. Computational Studies of 4 Structurally Related 2’,4’-diOH Chalcones to Examine their Antimalarial Activity by Binding to Falcipain-2 by Miriam Rossi et al.
The authors have provided revised manuscript which has improved and can be accepted for publication in Molecules.